# Wind power density in areas of Northeastern Brazil from Regional Climate Models for a recent past

**Augusto de Rubim Costa Gurgel** [1]*, **Domingo Cassain Sales**[2], **Kellen Carla Lima**[3]

1 Postgraduate Program in Climate Science, Center for Exact and Earth Sciences, Federal University of Rio Grande do Norte, Natal, Rio Grande do Norte, Brazil, 2 Fundação Cearense de Meteorologia e Recursos Hídricos, Fortaleza, Ceará, Brazil, 3 Escola de Ciências e Tecnologia, University of Rio Grande do Norte, Natal, Rio Grande do Norte, Brazil

☯ These authors contributed equally to this work.

* rubim10@yahoo.com.br

**Data Availability Statement:** The datasets analyzed during the current study are available from the following sources: Xavier observed data can be accessed at https://utexas.app.box.com/v/Xavier-etal-IJOC-DATA. The ECMWF-ERA5

## Abstract

Investments in renewable energy sources are increasing in several countries, especially in wind energy, as a response to global climate change caused by the burning of fossil fuels for electricity generation. Thus, it is important to evaluate the Regional Climate Models that simulate wind speed and wind power density in promising areas for this type of energy generation with the least uncertainty in recent past, which is essential for the implementation of wind farms. Therefore, this research aims to calculate the wind power density from Regional Climate Models in areas at Northeast of Brazil from 1986 to 2005. Initially, the ECMWF-ERA5 reanalysis data was validated against observed data obtained from Xavier. The results were satisfactory, showing a strong correlation in areas of Ceará and Rio Grande do Norte (except during the SON season), and some differences in relation to the wind intensity registered by observed data, particularly during the JJA season. Then, the Regional Climate Models RegCM4.7, RCA4 and Remo2009 were validated against the ECMWF-ERA5 reanalysis data, with all models successfully representing the wind speed pattern, especially from December to May. Four specific areas in Northeast of Brazil were selected for further study. In these areas, the RCMs simulations were evaluated to identify the RCM with the best statistical indices and consequently the lowest associated uncertainty for each area. The selected RCMs were: RegCM4.7_HadGEM2 (northern coastal of Ceará and northern coastal of Rio Grande do Norte) and RCA4_Miroc (Borborema and Central Bahia). Finally, the wind power density was calculated from the selected RCM for each area. The northern regions of Rio Grande do Norte and Ceará exhibited the highest wind power density.

## Introduction

Understanding wind speed and its local variability is crucial for conducting economic evaluations of wind farm development projects [1]. Environmental concerns are growing due to the impacts of climate change resulting from the burning of fossil fuels for electricity generation.

reanalysis data is available at https://cds.climate. copernicus.eu/cdsapp#!/dataset/reanalysis-era5-single-levels-monthly-means?tab=form. Data from the CORDEX-CORE Regional Climate Models can be found at https://esgf-index1.ceda.ac.uk/search/cordex-ceda/. The topography data used to create Fig 1 with R software was obtained from https://csidotinfo.wordpress.com/data/srtm-90m-digital-elevation-database-v4-1/. The location data for the wind farm in NEB, also used in Fig 1 with R software, was sourced from https://dadosabertos.aneel.gov.br/dataset/siga-sistema-de-informacoes-de-geracao-da-aneel. Shapefiles used to create Figs 2 to 7 were obtained using the Cartopy library in Python, which imports data from https://www.naturalearthdata.com.

**Funding:** The authors would like to thank the UFRN for paying the fees for publishing this article.

**Competing interests:** The authors have declared that no competing interests exist.

Consequently, wind energy technologies are rapidly advancing as a clean energy source, assisting countries explore new alternatives for electricity production with lower Greenhouse Gases emissions (GHG) [2, 3].

In Brazil, electricity production from renewable sources exceeds that from non-renewable sources. According to the International Renewable Energy Agency [4], in 2021, 83.9% of the country's energy was generated from renewable sources. However, one of Brazil's main challenges is its dependence on hydroelectric power, which accounted for 53.55% of the country's energy production in 2021. The Northeast region of Brazil (NEB) experienced a severe drought starting in 2012 [5] due to a lack of rainfall, impacting hydroelectric energy production and necessitating increased use of the thermal power plants. As a result, diversification in energy production is necessary, and wind energy has emerged as a promising alternative in Brazil, reflecting global trends. Over the past five years, wind energy production in Brazil has surged from 33,240.1 GWh in 2016 to 71,499.9 GWh in 2021. In 2021, wind energy accounted for 11.22% of the country's total energy production [4].

The NEB stands out as the country's prime location for wind energy development due to its strategic geographical positioning. This area benefits from intense, stable trade winds that blow consistently in the same direction [6]. The Atlantic Ocean coastline, along all states in this region, enhances wind speeds by minimizing obstructive barriers. Notably, the states of Ceará, Rio Grande do Norte and Bahia are the top producers of wind energy nationwide [7].

Wind speed can be measured by anemometers at various heights; however, measurements above 10m are rarely available due to the substantial investments required for structural reasons [8]. Mathematical methods, such as logarithmic and exponential functions, are useful for estimating wind speed at different heights [9, 10]. Furthermore, weather stations often exhibit measurements inaccuracies and significant spacing between them, resulting in low-quality data that may hinder the feasibility of wind projects [11].

In the face of these challenges, researchers use climatic reanalysis products to study wind speed in both past and present. Reanalysis involves assimilating historical time series with control models based on meteorological variables derived from observational data sources [12]. Recently, the European Centre for Medium-Range Weather Forecast (ECMWF) presented the ECMWF-ERA5 reanalysis dataset, which integrates extensive historical observations into global estimates. This dataset covers the period from 1950 to present, featuring a regular grid resolution of $0.25° \times 0.25°$ (latitude x longitude) and includes 137 levels of vertical surface pressure [13]. Many studies utilize reanalysis data to validate observed data in climate change studies [11, 14, 15].

In the field of climate change, the Coordinated Regional Climate Downscaling Experiment (CORDEX-CORE) [16] provides data from research centers that use Regional Climate Models (RCMs) for future climate projections. These simulations encompass the recent past (historical experiment) and extend to future projections up to 2100. RCMs are driven by boundary conditions from Global Climate Models (GCM) within the Coupled Model Intercomparison Project–Phase 5 (CMIP5) [17]. Through dynamic downscaling techniques, RCMs can capture regional climatic characteristics, such as extreme weather events, seasonal variations in precipitation, orographic precipitation and simulate features of regional-scale climatic anomalies such as the El Niño—Southern Oscillation (ENSO) [18].

Validating reanalysis data against observed data is crucial for assessing their accuracy. However, due to the lack of interpolated data, these validations are often conducted at specific points. Additionally, validating models using historical experiment is essential for improving future climate simulations. Nevertheless, few studies have validated wind speed and wind power density using CORDEX-CORE climate models and the Representative Concentration Pathways (RCP) for the NEB and for South America as a whole.

For instance, in South America, [10] used ECMWF-ERA-Interim reanalysis data to validate the RCM RegCM4 and evaluate wind speed projections for the near future (2020–2050) and distant future (2070–2098) under the RCP8.5 scenario, at 10 and 100 m height. The authors observed a decrease in wind speed in regions in the Northeast and Midwest regions of Brazil during the austral summer, along with an increase in this variable across all regions of Brazil during other seasons of the year, both for the near and distant future.

The contribution of this study is there are no previous studies that have validated the ECMWF-ERA5 reanalysis for the NEB with spatial wind speed variability. Additionally, some previous studies in this region focusing on wind speed used the ERA-Interim reanalysis, which exhibits statistically inferior metrics compared to the ECMWF-ERA5 reanalysis [19–21]. Furthermore, few prior studies on wind speed in the region of interest have utilized RCMs, and when utilized, typically only the RegCM4 model is considered, while the RCA4 and Remo2009 models are less commonly employed [10]. Finally, given that the NEB is the largest producer of wind energy in South America, a comprehensive study using RCMs to assess wind power density in this region is essential.

Thus, the general objective of the research is to calculate wind power density using Regional Climate Models in areas of Northeast Brazil for the recent past period from 1986 to 2005. Specifically, the aims are: (i) to spatially validate the ECMWF-ERA5 reanalysis for the NEB by comparing it with observed Xavier data; (ii) to validate the RCMs for the NEB relative to the reanalysis; (iii) to identify the RCMs with the best statistical indices for the NEB areas.

The next section describes the study area, the observed data, reanalysis and RCMs used for the validations, as well as the methodology applied to calculate the wind power density. Then, the results of the reanalysis and RCM validations, wind variability in NEB areas, and the selection of the model with the best metrics for calculating wind power density are presented. Finally, discussions and conclusions are provided.

## Materials

### Study area

The NEB region is situated between the meridians 48˚05' W and 35˚02' W and the parallels 1˚S and 18˚05'S. The region comprises nine Brazilian states: Maranhão (MA), Piauí (PI), Ceará (CE), Rio Grande do Norte (RN), Paraíba (PB), Pernambuco (PE), Alagoas (AL), Sergipe (SE) and Bahia (BA) (Fig 1). Four specific study areas were chosen in the NEB region, considering the number of installed wind farms and the varying climatic aspects. These study areas include: N-CE (orange): the northern coastal region of Ceará; N-RN (black): the northern coastal region of Rio Grande do Norte; Borborema (red): the region of the Borborema plateau that covers the states of Rio Grande do Norte, Paraíba and the wild landscape of Pernambuco, and finally, C-BA (green): the central region of Bahia.

The free software R was used to create the Fig 1, utilizing the libraries: geobr, ggspatial, lattice, ggplot2, raster and sp. The topography data was obtained from: https://csidotinfo. wordpress.com/data/srtm-90m-digital-elevation-database-v4-1/ and the location of active wind farms in the NEB was obtained from: https://dadosabertos.aneel.gov.br/dataset/siga-sistema-de-informacoes-de-geracao-da-aneel (free access).

### Data

**Xavier observed data.** The monthly observational database selected to validate the wind speed of the ECMWF-ERA5 reanalysis was provided by [22], which is referred to here as observed data. This database contains multiple atmospheric variables such as precipitation, evapotranspiration, maximum and minimum temperature, solar radiation, relative humidity,

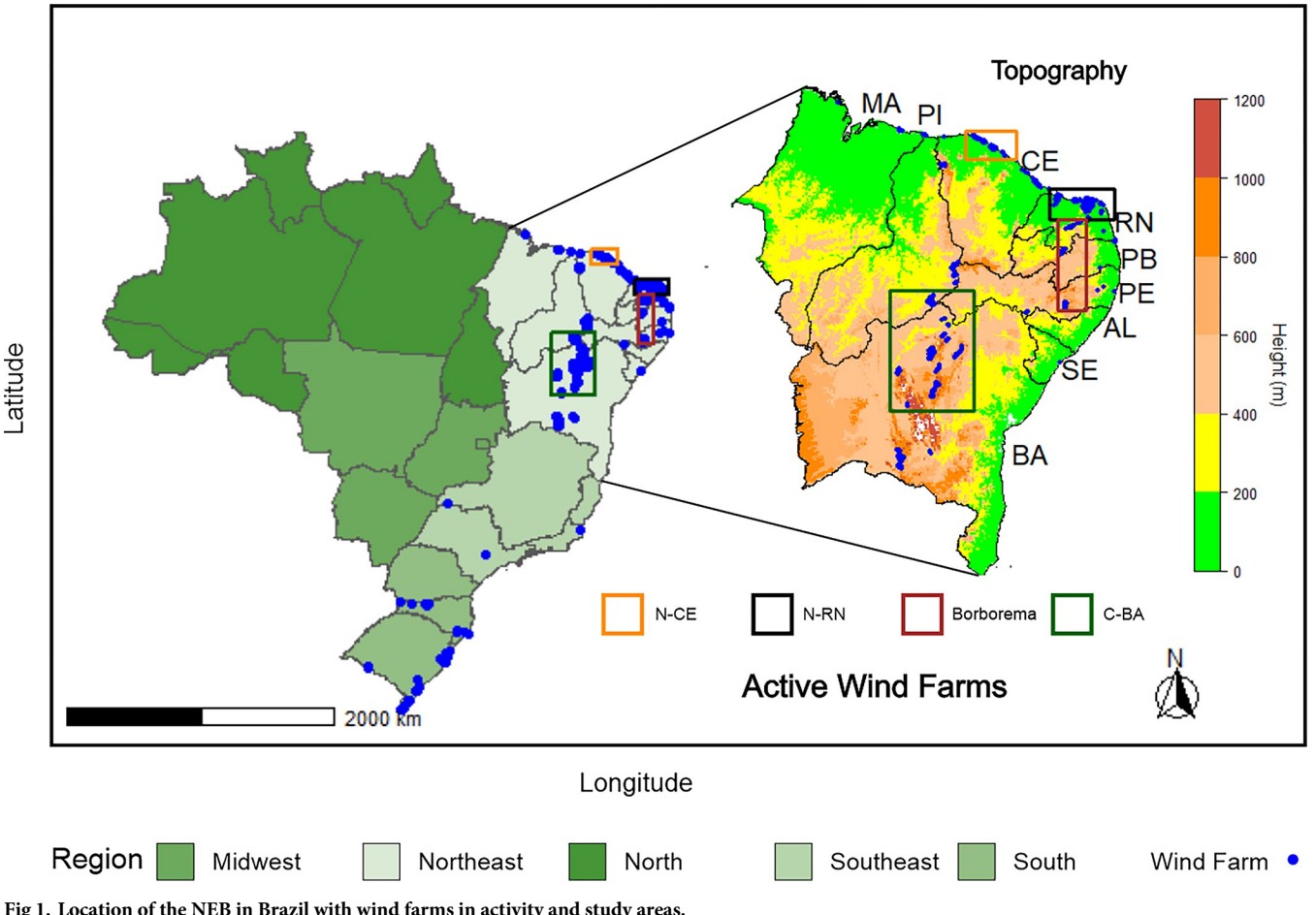

**Fig 1. Location of the NEB in Brazil with wind farms in activity and study areas.**

and wind speed. The database covers the period from 1980 to 2013. Six methodologies were employed to interpolate atmospheric variables, with the Inverse Distance Weighting (IDW) method emerging as the most effective for wind speed interpolation. The interpolation utilized databases derived from 3,625 pluviometers and 735 meteorological stations, collected by the National Institute of Meteorology (INMET), the National Water Agency (ANA), and the São Paulo State Department of Water and Electric Energy (DAEE) [22].

The authors utilized wind speed data at a 2m height from a dataset covering Brazil with a horizontal resolution of 0.25˚×0.25˚ latitude-longitude for the period from 1986 to 2005. This high-resolution dataset was employed in the study, but the interpolation process itself was not conducted by the present authors. This data was extrapolated to a 10m height for comparison with the data from the ECMWF-ERA5 reanalysis. The observed data, interpolated within the same grid as the reanalysis, enabled spatial validation of the reanalysis in the NEB region. This ensured the reanalysis dataset, which offers extensive spatial and temporal coverage as well as wind speed data at various altitudes, could be effectively utilized throughout the study.

**ECMWF-ERA5 reanalysis data.**   The monthly ECMWF-ERA5 reanalysis data were available in the Copernicus Climate Data Store [13]. The ECMWF-ERA5 is a global reanalysis product generated by the European Centre for Medium-Range Weather Forecasts (ECMWF), which incorporates several enhancements compared to its predecessor, the ERA-Interim. In terms of spatial resolution, the ECMWF-ERA5 features a horizontal grid spacing of 31 km, an

improvement over ERA-Interim's 80km. Moreover, ECMWF-ERA5 includes an enhanced vertical resolution and hourly output, in contrast to the six-hourly output of the ERA-Interim. Additionally, the ECMWF-ERA5 also employs a 4D-Var data assimilation system, representing a significant advancement over ERA-Interim.

In terms of physical parameterizations, ECMWF-ERA5 integrates the HTESSEL land surface scheme, which has undergone notable improvements compared to ERA-Interim. These enhancements include better hydrological representation, the introduction of soil texture maps, and improved parameterization of snow cover. Furthermore, ECMWF-ERA5 benefits from ongoing advancements in assimilating all-weather conditions, including successful integration of microwave humidity data [13].

For this study, wind speed data at 10 m height were utilized to validate ECMWF-ERA5 performance against observed data. Subsequently, wind speed data at 100 m height were employed to validate the RCMs data outputs. Additionally, the variability of wind speed at 100 m height was assessed in the selected NEB study areas. All data are available monthly for the period from 1986 to 2005, with a regular horizontal spatial resolution of 0.25˚ in latitude and longitude. The horizontal spatial grid was maintained to ensure comparability with observed data. However, for a spatial analysis involving RCMs, the grid spacing was interpolated to match the horizontal resolution of RCMs (0.44˚ x 0.44˚).

**CORDEX-CORE Regional Climate Models.**   This study utilized three RCMs from CORDEX-CORE [16]. RegCM4.7 was nested using outputs from three GCMs (HadGEM2-ES, NorESM1-M and MPI-ESM-MR). RCA4 was nested using outputs from five GCMs (HadGEM2-ES, NorESM1-M, MPI-ESM-LR, EC-EARTH and Miroc5), while Remo2009 was nested only with MPI-ESM-LR GCM output. This combination of regional-global models resulted in nine sets of output data.

RegCM4 employs various parameterizations to simulate atmospheric and surface processes. The radiation scheme used is NCAR CCM3 [23]. The land surface model scheme employed in RegCM4 is the Biosphere-Atmosphere Transfer Scheme (BATS) [24], and the cloud microphysics description is based on the Integrated Forecast System (IFS) from the ECMWF [25–27]. Furthermore, the planetary boundary layer scheme used in the RegCM4 is the Holtslag planetary boundary layer scheme [28].

Regarding the RCA4 model, the radiation parameterization is based on the HIRLAM scheme, originally developed for numerical weather prediction purposes [29]. For the surface, the RCA4 model utilizes the BATS surface scheme [30], while the surface scheme used in the model is referred to as the Land-Surface Scheme (LSS) and is classified as a second generation of LSSs [31]. In the Remo2009 model, the physical radiation parameterization is based on the Morcrette radiation scheme, as implemented in the ECHAM4 general circulation model [32]. Additionally, the model integrates a cloud microphysics scheme named PCI [33]. Furthermore, a global dataset of land surface parameters (LSP) has been developed [34].

The wind speed data at 100 m height from the RCMs were validated against ECMWF-ERA5 reanalysis data. These data have a monthly frequency and cover the recent past period from 1986 to 2005 and employing a horizontal spatial grid with a resolution of 0.44˚ in both latitude and longitude. Subsequently, the wind speed data from the RCMs at 100 m height were compared in the four selected study areas in the NEB for the recent past period (1986–2005), aiming to evaluate the models with the best statistical indices in each study area. Following this assessment, data from the model that showed the least uncertainty in simulating wind speed at 100 m height were utilized for further analysis of wind speed and calculation of wind power density during the recent past period (1986–2005).

The RCM RegCM4.7 features horizontal spatial resolution of 0.22˚ in latitude and longitude, whereas the RCA4 and Remo2009 models utilize a grid with a regular horizontal spatial

Table 1. RCMs belonging to CORDEX-CORE and their global forcings.

| RCM·(Country) | Resolution | GCM·(Country) | Resolution |
|---|---|---|---|
| RegCM4.7·(Italy) [35] | 0,22˚·×·0,22˚ | HadGEM2-ES·(UK)·[38] | 1,2˚·×·1,9˚ |
| | | NorESM1-M·(Norway)·[39] | 2,5˚·×·1,9˚ |
| | | MPI-ESM-MR·(Germany)·[40] | 1,9˚·×·1,9˚ |
| RCA4·(Sweden)·[36] | 0,44˚·×·0,44˚ | HadGEM2-ES·(UK)·[38] | 1,2˚·×·1,9˚ |
| | | NorESM1-M·(Norway)·[39] | 2,5˚·×·1,9˚ |
| | | MPI-ESM-LR·(Germany)·[40] | 1,9˚·×·1,9˚ |
| | | EC-EARTH·(International)·[41] | 1,1˚·×·1,1˚ |
| | | Miroc5·(Japan)·[42] | 1,4˚·×·1,4˚ |
| Remo2009·Germany)·[37] | 0,44˚·×·0,44˚ | MPI-ESM-LR·(Germany)·[40] | 1,9˚·×·1,9˚ |

resolution of 0.44˚ in latitude and longitude. Wind speed data from the RCA4 and Remo2009 models are initially available at 10m height. Thus, for the purpose of model comparison, RegCM4.7 data were interpolated to a horizontal spatial resolution grid of 0.44˚ in latitude and longitude, and the RCA4 and Remo 2009 data were extrapolated to a height of 100 m Table 1 provides details of the RCMs used, as well as the corresponding global models employed as large-scale forcings.

## Methodology

**Extrapolation of wind speed to 10m and 100m in the comparison between databases.** To ensure consistency in height for comparison, wind speed data from observed sources was extrapolated from 2 to 10 meters to match the height of ECMWF-ERA5 reanalysis data. Additionally, data from the RCA4 and Remo2009 models, initially available at 10 meters height, were extrapolated to 100 meters for direct comparison with ECMWF-ERA5. The extrapolation followed the logarithmic law expressed [43]:

$$U(z) = U(zr) \frac{ln\left(\frac{z}{z_0}\right)}{ln\left(\frac{z_r}{z_0}\right)} \tag{1}$$

Where: U(z) = wind speed at the desired height (m/s), $U(Z_r)$ = wind speed measured at the reference point (m/s), $Z_0$ = surface roughness length, Z = desired height, $Z_r$ = the reference height (ex: meteorological station). Surface roughness for each grid point in the models was estimated using values provided in the RegCM4 manual [44]. Eq (1) is appropriate for neutral stability conditions of the atmosphere. In the NEB, sea winds are expected to present neutral to unstable atmospheric conditions, as demonstrated by climatological data [45] and radiosonde observations [46]. Furthermore, previous study [10], the equation was employed without separating stability conditions because the study dealt with monthly data and a climate scale, rather than at a micrometeorology scale, similar to the situation in this study.

**Validations: Reanalysis versus observation and Regional Climate Models versus reanalysis across the NEB.** To assess the accuracy of the ECMWF-ERA5 reanalysis compared to observed data in the NEB for the recent past period (1986–2005), analyses were conducted using bias and Pearson's correlation coefficient. This validation was performed seasonally (with 60 values for each season, covering 20-year period) using wind speed data at 10 m height. Following this, the seasonal variability of wind speed at 100 m height throughout the NEB was analyzed for reanalysis data. The RCMs' data were validated by bias, identifying any seasonal

trends relative to the reanalysis. The bias is expressed by:

$$bias = \frac{1}{n} \sum\nolimits_{i=1}^{N} P_i \qquad (2)$$

Where, $P_i = P_{i,mod} - P_{i,obs}$ is the difference between the wind speed value of the ECMWF-ERA5 reanalysis ($P_{i,mod}$) and the wind speed value from the observed data $P_{i,obs}$ at the same location and time ($i = 1,2,3\ldots,n$), and n is the number of observations (with 60 values for each season covering the 20-years period). After calculating the bias, Student's t-tests were conducted to determine the statistical significance of the results. A 95% confidence interval and two-tailed distribution were assumed [47].

$$t = \frac{\bar{P}_{obs} - \bar{P}_{mod}}{\sqrt{\left( \left( \frac{s_{obs}^2}{n_{obs}} \right) + \frac{s_{mod}^2}{n_{mod}} \right)}} \qquad (3)$$

Which: $\bar{P}_{obs}$, $\bar{P}_{mod}$ it's the averages; $S_{obs}$ and $S_{mod}$ are the standard deviations; and $n_{obs}$ and $n_{mod}$ are the number of elements in the time series. The "obs" index represents the observed data and the "mod" represents the reanalysis data.

Pearson's correlation coefficient describes the degree of collinearity between simulated and observed data. This index ranges from -1 to 1, with the best results close to value 1 [48]. The coefficient is mathematically expressed by:

$$r = \frac{\sum_{i=1}^{n} (P_{obs,i} - \bar{P}_{obs})(P_{mod,i} - \bar{P}_{mod})}{\sqrt{\left( \sum_{i=1}^{n} P_{obs,i} - \bar{P}_{obs} \right)^2 x \left( \sum_{i=1}^{n} P_{mod,i} - \bar{P}_{mod} \right)^2}} \qquad (4)$$

Where: $P_{obs,i}$ is the observed data value at time i; $\bar{P}_{obs}$ is the average observed data value; $P_{mod,i}$ is the estimated value of the reanalysis at time i; $\bar{P}_{mod}$ is the estimated mean value of the reanalysis; n is the number of observations; r is the correlation coefficient. The result was considering statistical significance for p-valor<0.05.

**Monthly variability of wind speed at 100 m from the ground in areas of the NEB.** To thoroughly understand the wind behavior in each study area of the NEB during the recent past (1986–2005), a descriptive numerical analysis of wind speed intensity at 100m height was conducted using ECMWF-ERA5 reanalysis data. This analysis involved constructing boxplots to examine monthly variations in wind speed, capturing measures such as minimum recorded, 1st quartile, median, mean, 3rd quartile and maximum wind speeds. Additionally, variance and standard deviation were calculated and included to provide a comprehensive understanding of wind speed variability.

**Evaluation: Regional Climate Models versus reanalysis in NEB areas.** RCMs were individually validated across the NEB region against ECMWF-ERA5 reanalysis data. However. The NEB region is extensive and exhibits several active meteorological phenomena contributing to its climate diversity. Given the high horizontal spatial resolution of RCMs, certain models may yield superior results for specific areas [18]. Therefore, it is crucial to evaluate which models provide the best statistical indices for simulating wind speeds compared to the reanalysis, particularly in areas with a significant number of installed wind farms. This evaluation process facilitates the calculation of wind power density in these areas with reduced uncertainty [18].

The annual wind speed cycle between reanalysis and RCMs was evaluated for the study areas in the NEB region to determine models' ability to capture the wind speed patterns observed in the reanalysis data. According to [18], one method to evaluate the improvements

or additional insights provided by RCMs is using Taylor diagram. The Taylor diagram [49] presents robust statistical metrics in a single diagram, including the Root of Mean Square Error (RMSE), Pearson's Correlation Coefficient, and standard deviation. The outcomes of these metrics indicate the performance of the models. The fundamental dataset for comparison consists of the wind speed data from the reanalysis. In the Taylor diagram, a model's proximity to the reanalysis indicates better simulation performance for the specific area. The RMSE, defined mathematically for use in Talor Diagram, is as follows:

$$RMSE = \sqrt{S(mod) + S(obs) - 2*S(mod)*S(obs)*r(mod, obs)} \tag{5}$$

Where: S(mod) and S(obs) is the standard derivation and r(mod,obs) is the Pearson's correlation coefficient between RCM and ECMWF-ERA5 reanalysis. The RMSE ranges from 0 to infinity, with 0 being the ideal result. Here "mod" is attributed to RCMs and "obs" is attributed to ECMWF-ERA5 reanalysis data.

The standard deviation of the model should be the closest to the value of the observation. The standard deviation is calculated:

$$S = \sqrt{\frac{\left(P_{data,i} - \bar{P}_{data}\right)^2}{(n-1)}} \tag{6}$$

Where: $P_{data,i}$ is the simulated value of the ECMWF-ERA5 reanalysis/RCM at time i; $\bar{P}_{data}$ is the average simulated value of the ECMWF-ERA5 reanalysis/RCM; n is the number of observations; S is the standard deviation. Eq (6) is used twice: first to calculate the standard deviation for the ECMWF-ERA5 reanalysis, and then the same formula is used to calculate the standard deviation for the RCMs in the Taylor diagram.

**Wind power density from Regional Climate Models in NEB areas.** Wind power density estimates the amount of electrical energy produced per unit square meter (W/m$^2$) in a specific region. In this study, wind power density calculations were based on monthly data of air density and wind speed [51]. The wind speed variable is extremely important in these calculations, as even a small inaccuracy can generate a significant error due to its cubed relationship. To reduce this uncertainty, some studies employ ensembles averaging of models [10, 50]. In our research, wind speed data from the best-performing model for each study area was utilized to achieve the most accurate results possible. The mathematical equation used is [51]:

$$P(v) = \frac{1}{2} \times \rho v^3 \tag{7}$$

Where: ρ = air density; v = wind speed (m/s).
The air density, can be defined as [43]:

$$\rho = \frac{\left(353,4\left(1 - \frac{z}{45271}\right)^{5,2624}\right)}{273,15 + T} \tag{8}$$

Where: Z = height (m), T = air temperature (˚C)
The temperature was obtained for each CORDEX-CORE model (RegCM4.7, RCA4 and Remo2009) at 2 m height. To use these values at 100 m height, a dry adiabatic laps rate was considered, in which every 100 m of height the temperature decreases by 1˚C [52].

## Results

The Figs from 2 to 7 were created using the free software Python, with cartopy library, which imports data from NaturalEarthFeature. This data is freely available from: http://www.naturalearthdata.com. Figs 8–11 were created using the free software R, without the need for any specific library.

### Validation: Reanalysis versus observation across the NEB

Fig 2 illustrates the seasonal variability of Pearson's correlation coefficient (left) and bias (right) of wind speed intensity from the reanalysis versus observed data throughout the NEB for the recent past from 1986 to 2005.

During the austral summer (DJF), Pearson's correlation coefficient (Fig 2A) shows that the northern region of the NEB exhibits a very strong correlation, predominantly ranging from 0.9 and 1.0. NEB's western sector has the lowest values of Pearson's correlation coefficient, considered very weak to weak, with values up to 0.4. The southern region of the NEB displays a Pearson's correlation coefficient ranging from very weak to strong, with values above 0.0 to

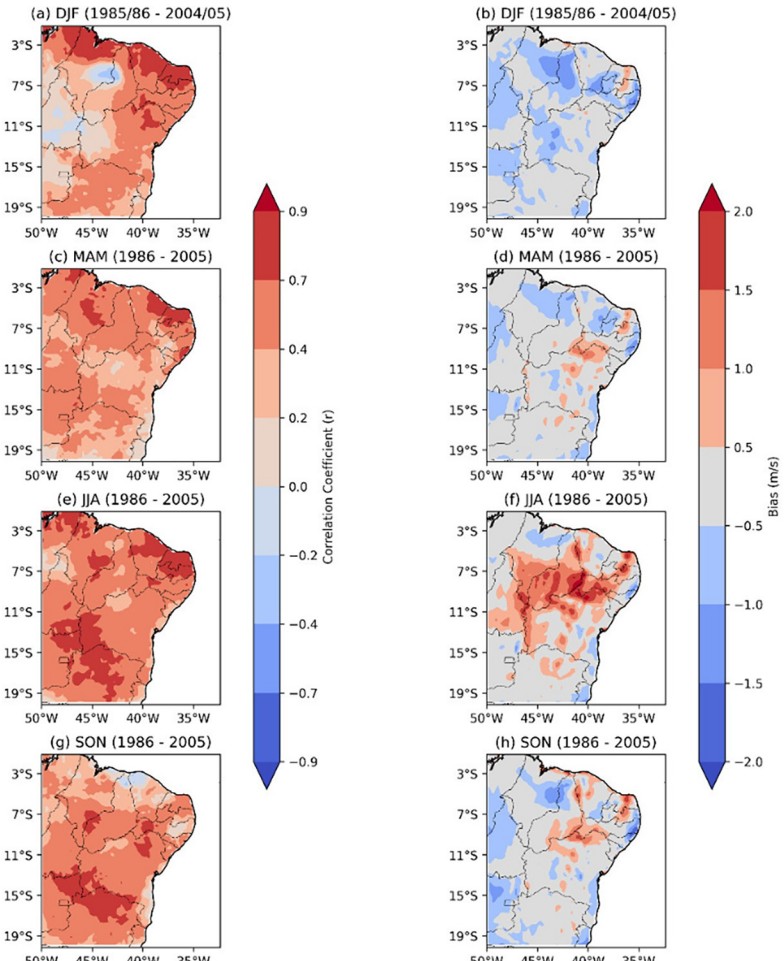

**Fig 2. Seasonal variability of Pearson's correlation and bias of wind speed, reanalysis versus observed data.** (a) Correlation DJF, (b) bias DJF, (c) correlation MAM, (d) bias MAM, (e) correlation JJA, (f) bias JJA, (g) correlation SON, (h) bias SON.

0.7. For the bias results during this season (Fig 2B), the reanalysis shows absolute values lower than 0.5m/s in most of the NEB, with a tendency to underestimate the region by up to 1m/s.

In the subsequent quarter, the austral autumn (MAM–Fig 2C), there is a reduction in Pearson's correlation coefficients. In the northern region of the NEB, Pearson's correlation coefficient ranges from 0.4 to 0.9, indicating moderate to strong correlation. The central and southern regions of the NEB predominantly exhibit Pearson's correlation coefficient between 0.2 and 0.7, i.e., indicating weak to moderate correlation. Regarding the bias (Fig 2D), most of the NEB presents values in the range between -0.5 and 0.5m/s, with a decreasing tendency in the regions previously underestimated.

The season with the highest values of Pearson's correlation coefficient is the austral winter (JJA–Fig 2E), during which nearly all areas of the NEB exhibit values ranging from moderate to strong (0.4 to 0.9). Coastal areas in the northern part of the NEB (CE and RN) even reach the classification of very strong (0.9 to 1). Conversely, the wind speed bias analysis (Fig 2F) reveals that austral winter is the season with the largest spatial extent of wind speed overestimation, with values reaching up to 2 m/s in the central area of the NEB.

Finally, in the austral spring (SON–Fig 2G), Pearson's correlation coefficient values are the lowest for the NEB, ranging from 0.0 and 0.4, which is considered very weak to weak along northern coast of the NEB. However, some areas in the southern and central NEB show Pearson's correlation coefficients ranging from moderate to strong (0.4 to 0.9). The bias analysis (Fig 2H) indicates a reduction in the spatial extent of wind speed overestimation in the central NEB. All results are statistically significant based on the t-test with 95% confidence level for the bias results, and statistically significant for Pearson's correlation coefficient with a p value<0,05.

**Seasonal variability of wind speed at 100m height across the NEB.** Fig 3 presents the seasonal mean wind speed at 100 m height for the ECMWF-ERA5 reanalysis during the recent

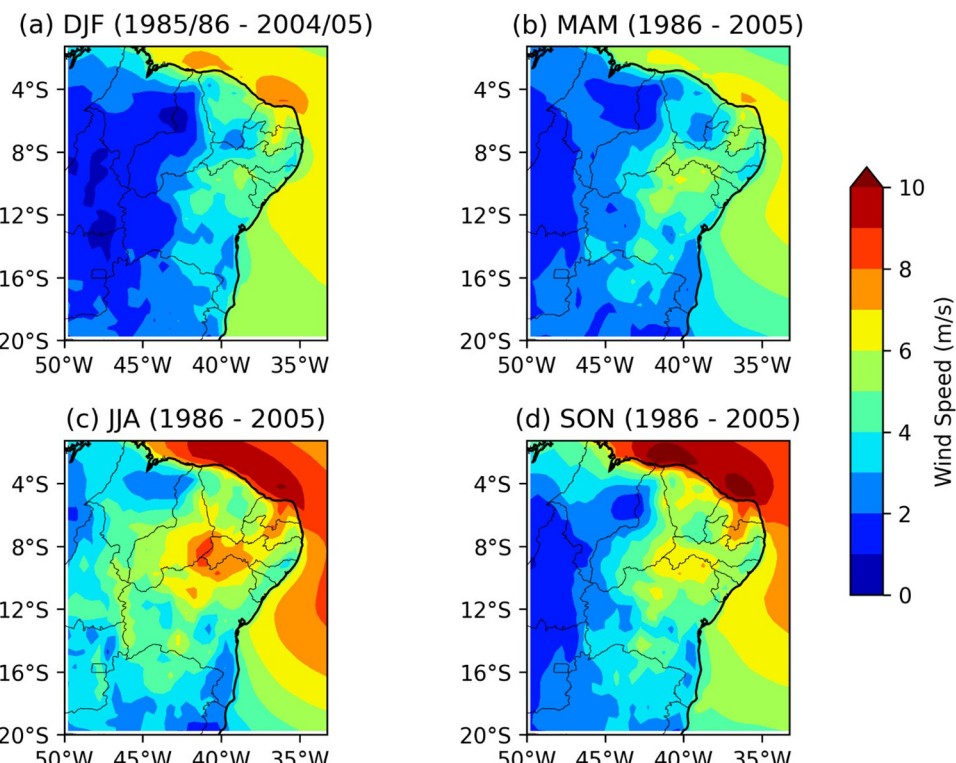

**Fig 3. Seasonal average wind speed (m/s) at 100m from the reanalysis for the recent past (1986–2005).** (a) DJF, (b) MAM, (c) JJA e (d) SON.

past period from 1986 to 2005. Climatologically, the highest wind speeds are observed in the JJA and SON seasons (Fig 3C and 3D, respectively), with mean velocities ranging between 6 and 9 m/s in the northern coastal region, encompassing the states of MA to RN, as well as in the Borborema plateau region spanning the states of RN, PB, and PE, extending to the northern and central regions of BA.

During the austral summer (DJF–Fig 3A) and austral autumn (MAM–Fig 3B), the coastal regions from MA to RN, the Borborema plateau, and the northern part of BA, exhibit velocities ranging between 4 and 7 m/s. Although wind speeds in these seasons are lower compared to the JJA and SON quarters, they still meet the minimum requirements for wind energy production.

**Validation: Regional Climate Models versus reanalysis across the NEB.** The models used in this study are named based on the combination of RCM and GCM. The nomenclature on the left refers to the RCM, and on the right to the GCM that drives it globally. Thus, the models are named: RegCM4.7_HadGEM2, RCA4_HadGEM2, RegCM4.7_MPI, RCA4_MPI, Remo2009_MPI, RegCM4.7_NorESM, RCA4_NorESM, RCA4_Miroc, and RCA4_EARTH.

Fig 4 shows the bias of the wind speed intensity of the RCMs versus reanalysis data across the NEB for the austral summer (DJF) during the recent past period from 1986 to 2005. The

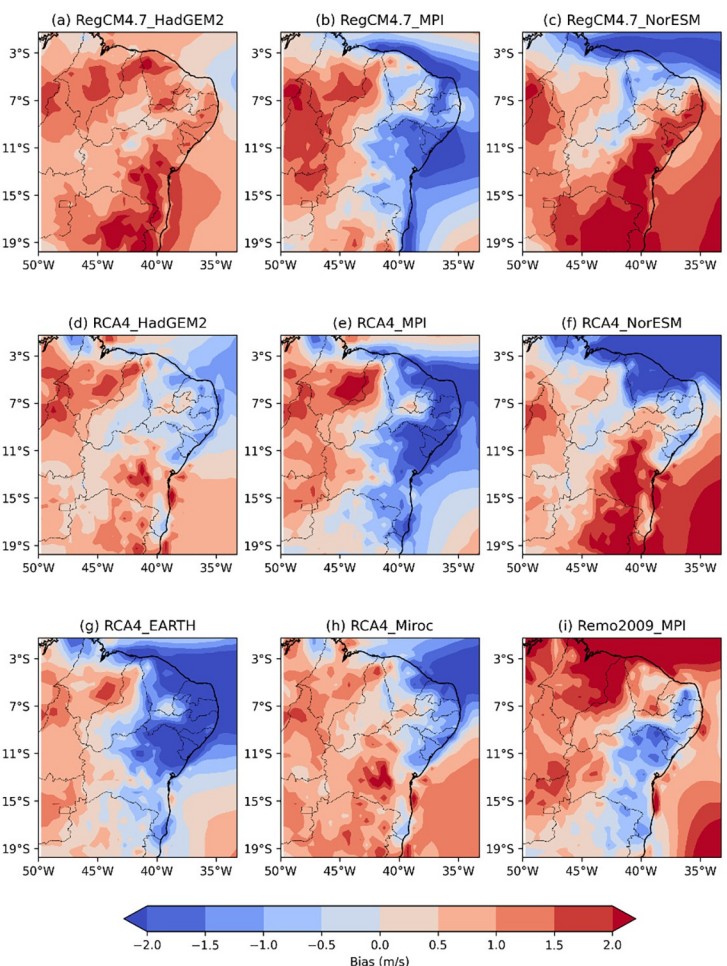

**Fig 4. Bias of RCM wind speed intensity versus reanalysis data across the NEB for the austral summer (DJF).**

RegCM4.7_HadGEM2 model (Fig 4A) is the only model that overestimates wind speed intensity across the NEB.

Models RegCM4.7_MPI, RCA4_HadGEM2, RCA4_MPI, RCA4_EARTH, RCA4_miroc, and Remo2009_MPI (Fig 4B, 4D, 4E and 4G–4I) display similar patterns in wind speed bias. They tend to overestimate wind speed in the states of MA and PI, with the models driven by MPI GCM showing more intense and widespread areas of overestimation. Additionally, these models tend to underestimate wind speed along the entire NEB coastline, except for the Remo2009_MPI, which underestimate only the eastern coastline.

The RegCM4.7_NorESM and RCA4_NorESM models (Fig 4C and 4F) exhibit similar patterns where they underestimate wind speed along the northern coast of the NEB and overestimate it along the eastern coast, extending into more central regions. The RegCM4.7_NorESM tends to have larger areas of wind speed overestimation, while the RCA4_NorESM has more areas of wind speed underestimation.

Fig 5 shows the wind speed intensity bias of the RCM versus reanalysis across the NEB for austral autumn (MAM) during the recent past period from 1986 to 2005. The bias pattern of wind speed simulated by the RCM is similar to the pattern observed in the DJF period.

The RegCM4.7_HadGEM2 model (Fig 5A) continues to overestimate wind speed across the NEB but shows a reduction in positive bias errors compared to the previous season, with some areas now exhibiting underestimated wind speed intensity.

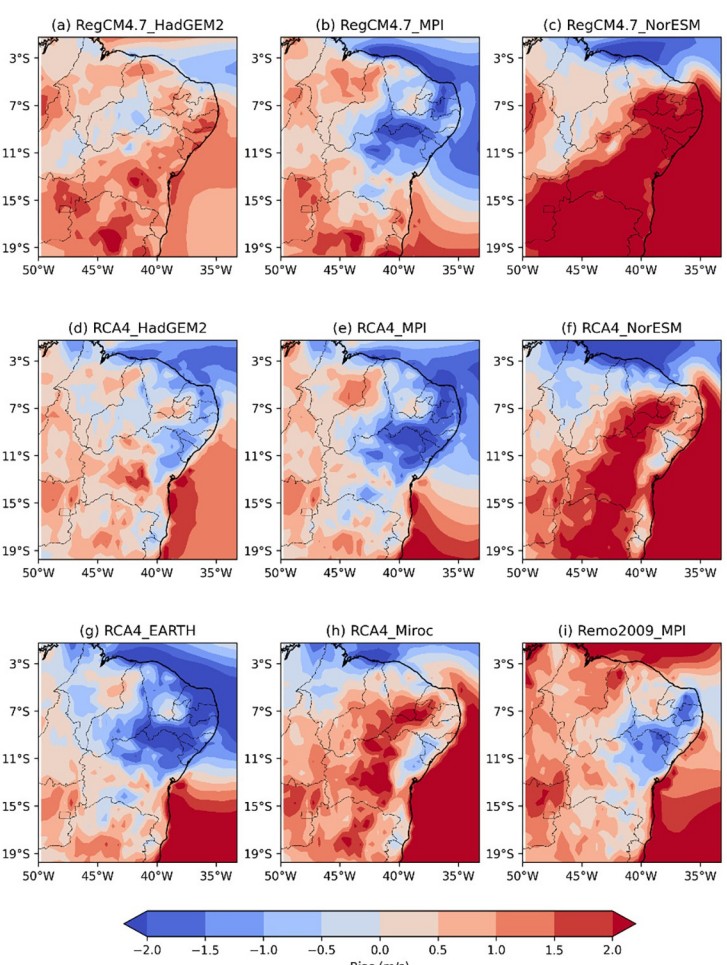

**Fig 5. Bias of RCM wind speed intensity versus reanalysis data across the NEB for the austral autumn (MAM).**

The RegCM4.7_MPI, RCA4_MPI, and Remo2009_MPI models (Fig 5B, 5E and 5I) exhibit reduced intensities and overestimated areas in MA and PI, particularly in the REMO2009_MPI model. The areas of underestimation in these models have decreased, with RegCM4.7_MPI and RCA4_MPI underestimating wind speed from the eastern part of the NEB to central BA. The RCA4_miroc model (Fig 5H) shows an intensification of overestimated wind speed in the central and southern regions of the NEB.

The RegCM4.7_NorESM and RCA4_NorESM models (Fig 5C and 5F) display an intensification of wind speed. The regions of wind underestimation are now limited to the states of MA, PI and CE. The RegCM4.7_NorESM model overestimates wind speed from PB to BA, while the RCA4_NorESM model concentrates this overestimation further west in PB extending to the southern part of BA.

Fig 6 shows the wind speed intensity bias of the RCMs versus reanalysis data across the NEB for the austral winter (JJA) during the recent past period from 1986 to 2005. The wind speed patterns captured by the simulations tend to be overestimated in all models using RCM RegCM4.7 and Remo2009_MPI (Fig 6A–6C and 6I) across the NEB. Simulations using the RCA4 model (Fig 6D–6H) capture similar wind speed patterns. The eastern coast of the NEB, from PB to BA is characterized by a slight underestimation of wind speed.

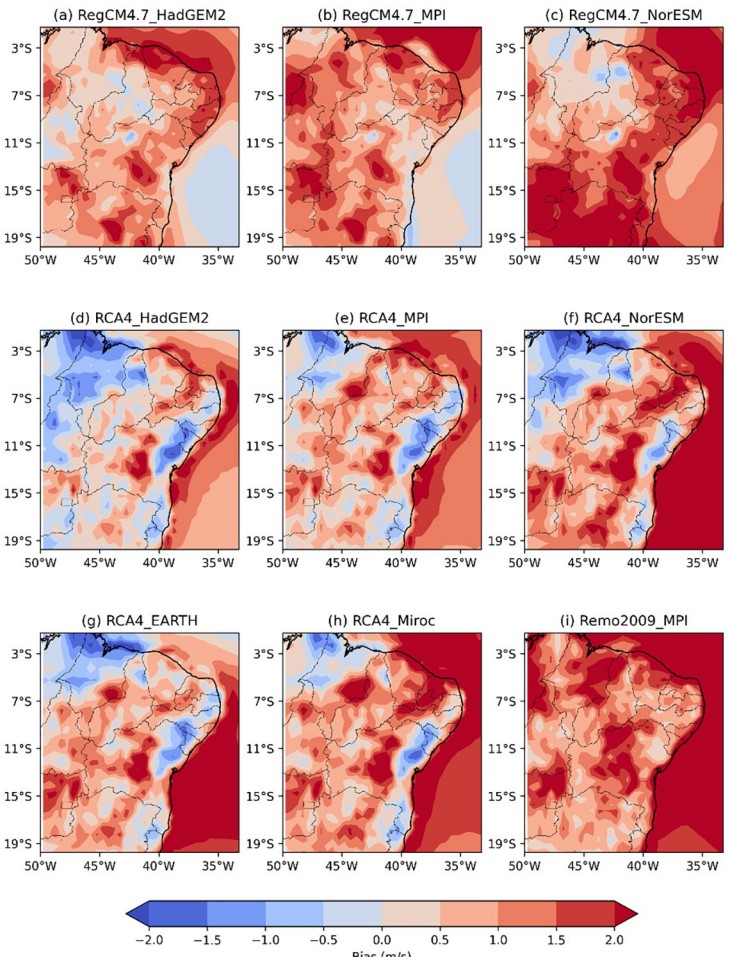

**Fig 6. Bias of RCM wind speed intensity versus reanalysis data across the NEB for the austral winter (JJA).**

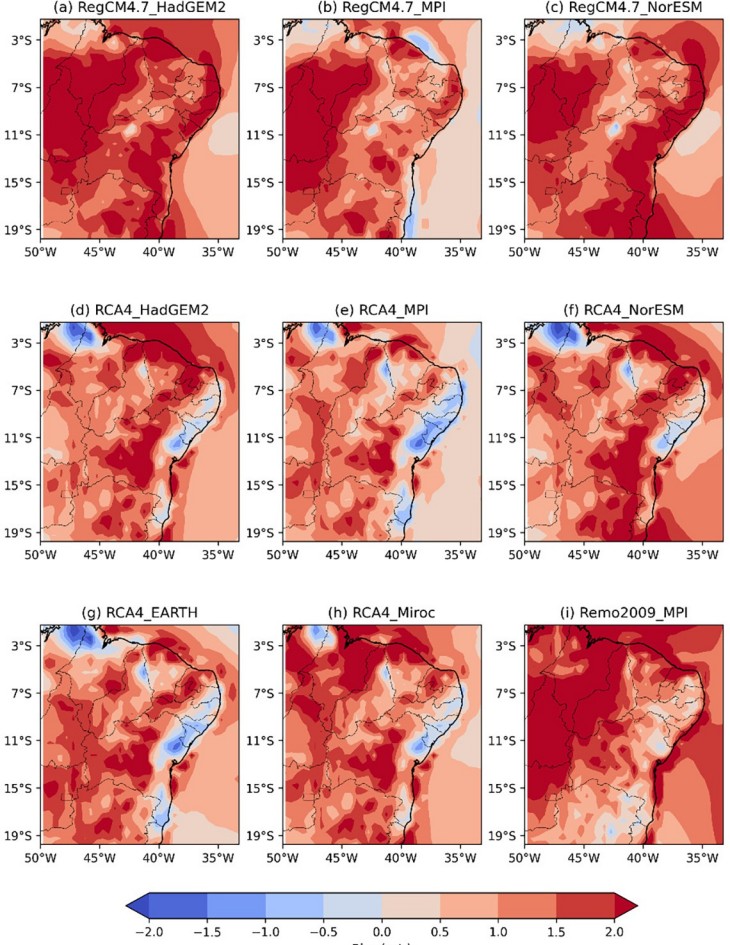

**Fig 7. Bias of RCM wind speed intensity versus reanalysis data across the NEB for austral autumn (SON).**

Fig 7 illustrates the wind speed intensity bias of the RCMs versus reanalysis data across the NEB for the austral spring (SON) during the recent past period from 1986 to 2005. The RCMs show an increase in wind speed across all regions of the NEB. In all models, both the spatial extend and intensity of overestimation are amplified. The eastern region of the NEB, from PB to central BA, continues to be represented similarly to the JJA period by simulations using the RCA4 model (Fig 7D, 7E, 7G and 7H) and Remo2009 (Fig 7I). The RegCM4.7_HadGEM2 model (Fig 7A) overestimates wind speeds throughout the NEB, with velocities exceeding 1 m/s and some regions showing values above 2 m/s.

Bias is a simple statistical metric calculated by subtracting the values between the model and the ECMWF-ERA5 reanalysis data. To enhance the accuracy of the results obtained, four study areas within the NEB region were selected (see areas in Fig 1). The monthly variability of wind speed in these areas will be analyzed using ECMWF-ERA5 reanalysis data. Subsequently, the models will be validated using the Taylor diagram.

## Monthly variability of wind speed at 100 m height in areas of the NEB

Fig 8 shows the monthly distribution of wind speed from the ECMWF-ERA5 reanalysis data at 100m height in the NEB study areas, as represented in Fig 1, for the recent past period from

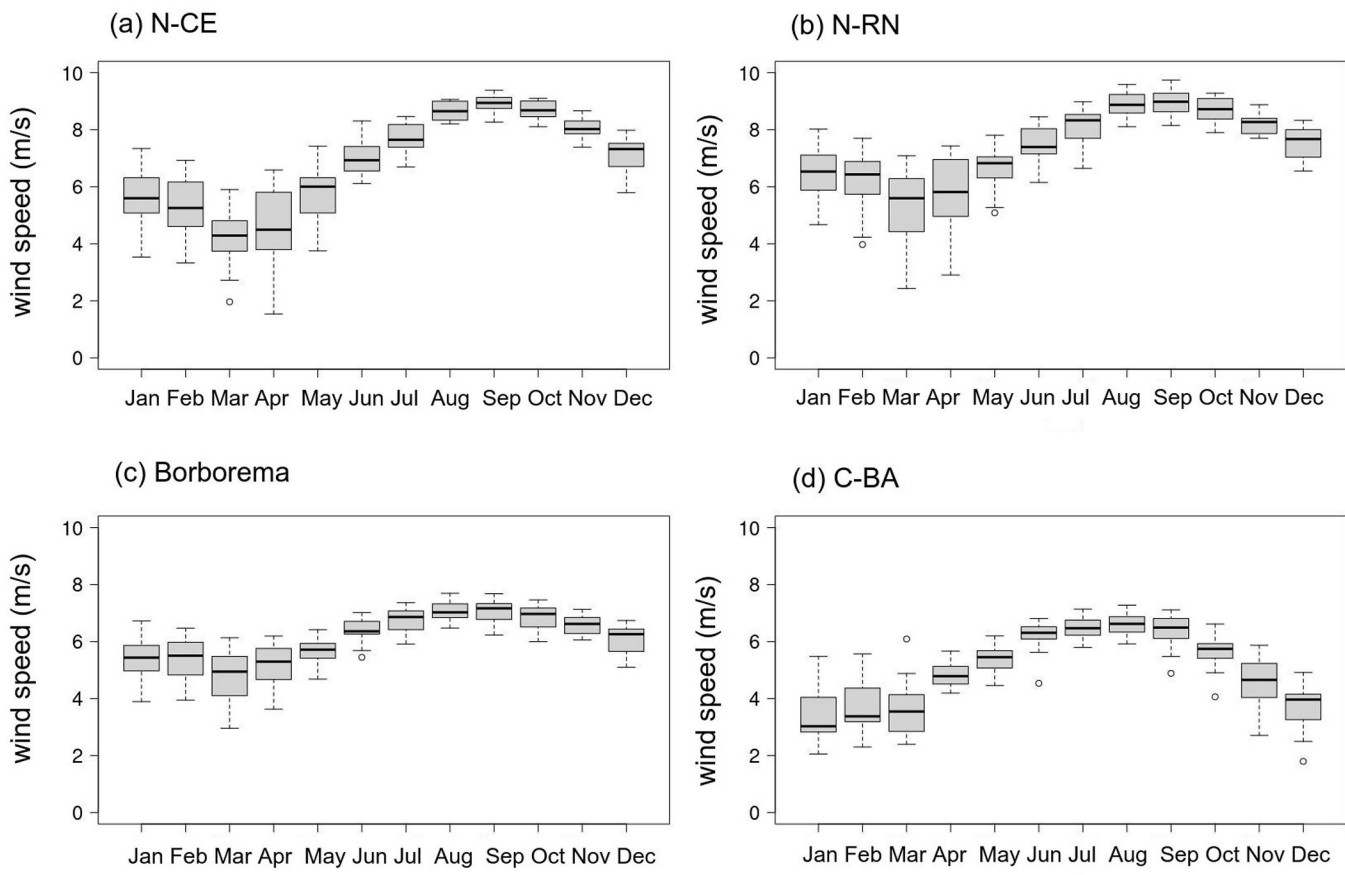

**Fig 8. Monthly variability of wind speed at 100m from the ground in NEB areas.** (a) northern Ceará (N-CE), (b) northern Rio Grande do Norte (N-RN), (c) Borborema (d) central Bahia (C-BA), based on the ECMWF-ERA5 reanalysis for the recent past period from 1986 to 2005.

1986 to 2005. All study areas have average wind speeds above 6 m/s (except C-BA), indicating potential for wind energy production [6]. The N-CE and N-RN areas (Fig 8A and 8B) are coastal, and thus exhibit the highest velocity averages, at 6.78 and 7.39 m/s, respectively. The highest velocities recorded for these areas occur during the austral spring, with values of 9.38 and 9.75 m/s, respectively. The N-CE area shows the highest wind speed variance, with a standard deviation of 1.77 m/s.

The Borborema area (Fig 8C) has an annual average wind speed of 6.0 m/s. The maximum wind speed is 7.59 m/s in the austral spring, while the minimum of 2.91 m/s in the austral autumn. The Borborema area exhibits the lowest standard deviation among all study areas, at 0.94 m/s. The C-BA area (Fig 8D) has an annual average wind speed of 5.04 m/s, with a maximum wind speed of 7.26 m/s recorded during the austral winter, while the austral summer register a minimum speed of 1.81 m/s.

## Evaluation: Regional Climate Models versus reanalysis in NEB areas

Fig 9 shows the monthly wind speed variability for each NEB study area. The models studied are able to capture the pattern of the annual wind cycle produced by the reanalysis, although they sometimes overestimate or underestimate wind speeds. Generally, the models underestimate wind speed during the first half of the year and overestimate it during the second half in

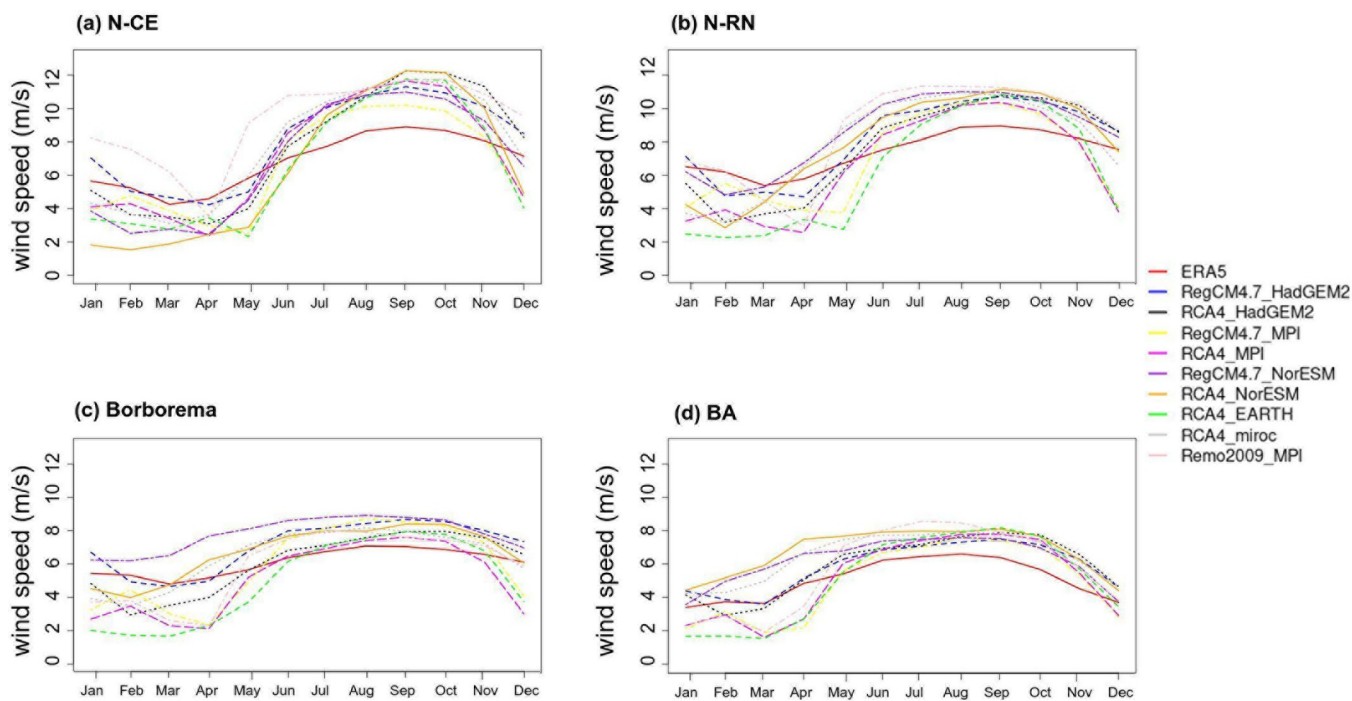

**Fig 9. Monthly wind speed variability (m/s) for reanalysis and RCMs for NEB areas (1986–2005).** a) N-CE, b) N-RN, c) Borborema and d) C-BA.

the N-CE and N-RN areas (Fig 9A and 9B). In the Borborema and C-BA areas (Fig 9C and 9D), underestimation periods are shorter, limited to the first four months of the year, with a tendency to overestimate wind speed for the remaining months compared to the reanalysis. Despite these errors presented, the models' ability to simulate the annual wind cycle in the areas suggests they can be reliably used for this research and for studies involving future projections.

Fig 10 shows the Taylor diagrams for the study areas in the NEB, comparing all models with each other within specific regions. The best models for each area are determined based on their highest correlation coefficient values and both lowest standard deviation and RMSE values when compared with the ECMWF-ERA5 reanalysis.

In the N-CE area (Fig 10A), the RegCM4.7_HadGEM2 model demonstrates the strongest correlation (>0.80), lowest standard deviation (2.50 m/s) and the second lowest RMSE (1.79 m/s) among the models. This model is identified as the most accurate for simulating wind speed in the N-CE area. Regarding the N-RN area (Fig 10B), most models show similar performance, though the RCA4_EARTH model stands apart from the other in the diagram. The RegCM4.7_HadGEM2 model exhibits a strong correlation (0.72), with RMSE and standard deviation values comparable to RegCM4.7_NorESM, establishing it as the optimal model for this area.

In the Borborema area (Fig 10C), the RCA4_Miroc model stands out with lowest RMSE value (1.29 m/s), strong correlation (0.70) and low standard deviation (1.74 m/s). This model is therefore recommended as the best representation of wind speed in the Borborema region. RegCM4.7_NorESM also performs well but with a slightly lower correlation of 0.54.

In the C-BA area (Fig 10D), three models show the most favorable statistical indices: RegCM4.7_NorESM, RCA4_miroc and RCA4_NorESM, all demonstrating strong Pearson's correlations (0.72, 0.77 and 0.72). Regarding the standard deviation, the RCA4_Miroc and RCA4_NorESM models presented similar values (1.55 m/s), and the RMSE presented by the

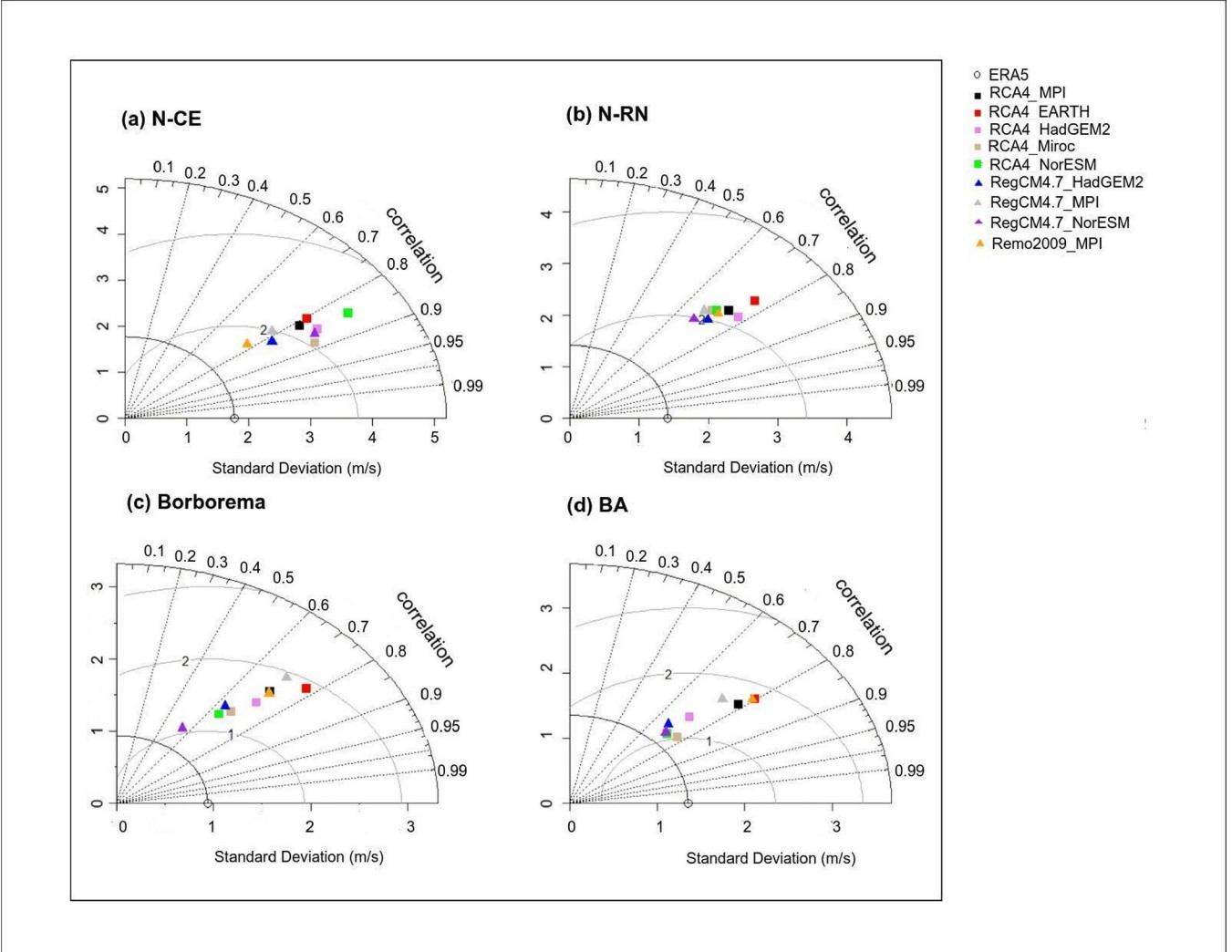

**Fig 10. Taylor diagram between reanalysis and RCM for NEB areas in the recent past between 1986 and 2005.** (a) N-CE area; (b) N-RN area; (c) Borborema area (d) C-BA area.

RCA4_Miroc was the lowest found (1.02 m/s). Thus, it was chosen in the C-BA area. These findings underscore the models' capability to replicate the annual wind cycle in these regions, supporting their suitability for current research and future projection studies.

## Wind power density from Regional Climate Models in NEB areas

Fig 11 shows the amount of wind power density for the NEB areas during the recent past (1986 to 2005) derived from the RCMs exhibiting the best statistical indices and lower uncertainties compared to the ECMWF-ERA5 reanalysis data. The area with the highest annual average energy density is the N-RN, recording 4.963,4 W/m$^2$ per year, followed closely by N-CE with 4.962,4W/m$^2$ per year. Borborema and central Bahia are rank third and fourth in potential, respectively, with 2.240,5 W/m$^2$ per year and 2.184,4 W/m$^2$ per year. These latter areas exhibit less than half the energy density of N-CE and N-RN.

The second half of the year shows elevated wind power density in NEB. Approximately 79,30% of the wind power density in the N-CE area occurs between July and December, with

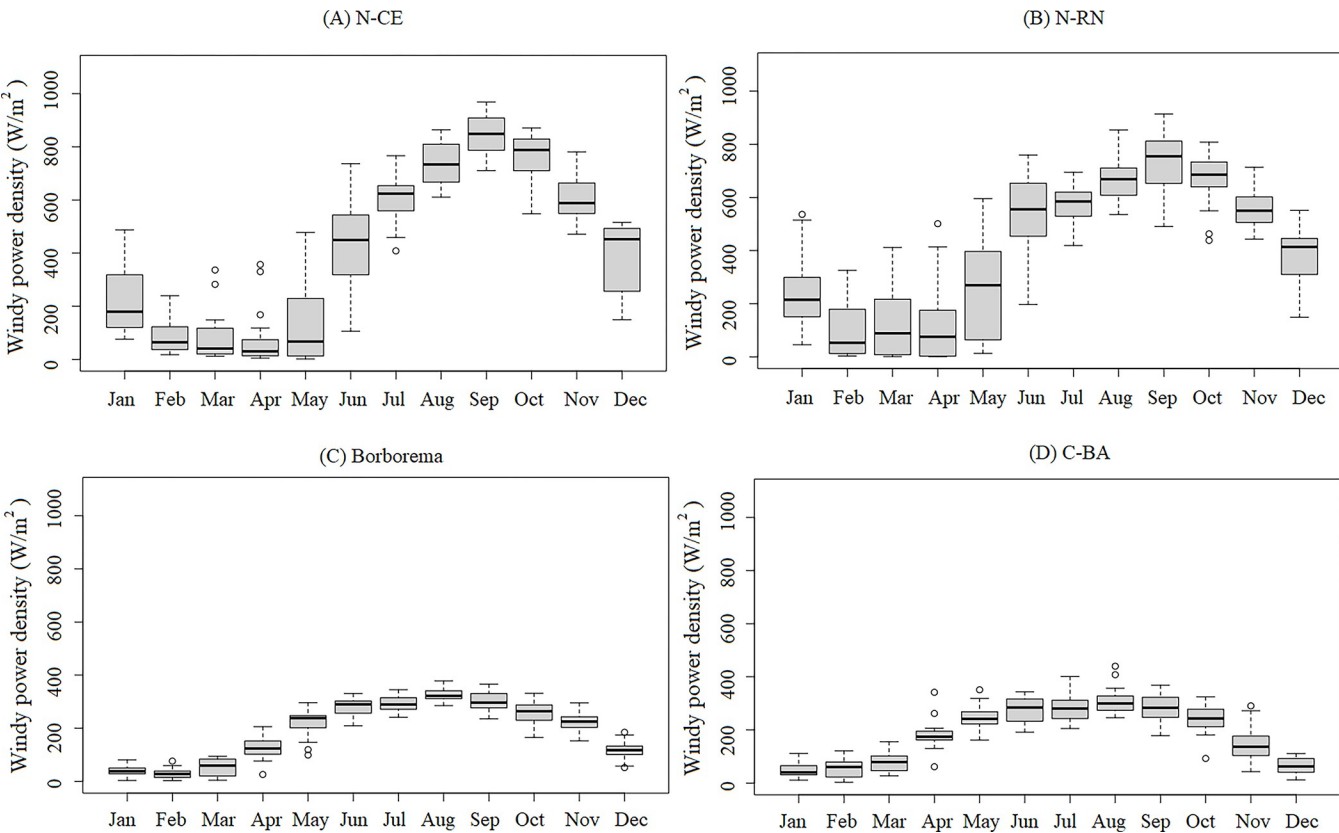

**Fig 11. The amount of wind power density for the NEB areas in the recent past (1986 to 2005).** (a) N-CE; (b) N-RN; (c) Borborema; (d) C-BA.

similar trends observed in other regions: 72.01% for N-RN, 66,91% for Borborema and 59.67% for C-BA. This phenomenon is largely attributed to the influence of the ITCZ and SASH weather systems, which enhance wind strengths during the second half of the year.

## Discussion

The utilization of reanalysis data to validate RCMs across various countries has become increasingly common in recent studies [10, 50, 53]. Typically, due to the limitations and significant gaps in meteorological station data (observed data), reanalysis data are used to enhance the accuracy of results in comparison to observed data [11]. This study employs the ECMWF-ERA5 reanalysis data due to its superior statistical metrics when compared to other reanalysis datasets (such as ERA-Interim, MERRA-2, and CFSv2) both in Brazil and globally [19–21].

Therefore, it is crucial to first assess the performance of the ECMWF-ERA5 reanalysis data relative to observed data on a seasonal basis. The quality of these data within Brazilian territory remains unknown [21], particularly for the NEB, which is characterized by high natural climatic variability [54]. This variability is influenced by atmospheric systems operating on different temporal and spatial scales, including the Intertropical Convergence Zone (ITCZ), High-Level Cyclonic Vortices (HLV), Frontal Systems (FS), Easterly Wave Disturbances (EWD) and South Atlantic Convergence Zone (SACZ) [55, 56]. These factors can complicate the accurate reproduction of specific wind patterns in this region.

Previous studies have evaluated reanalysis data in relation to point-based observed data. For example, in Bahia (BA), located in the NEB, [57] used monthly wind speed data from

November 2009 to December 2018 to compare reanalysis data with wind speed data measured at a 78m height on a wind tower. The ECMWF-ERA5 reanalysis demonstrated a correlation of 0.95 with the observed data, thereby validating the reanalysis data.

In this study, the comparison was conducted using observed data sourced from [22], who interpolated data from INMET using the Inverse Distance Weighting (IDW) methodology. Given the limited number of meteorological stations in the NEB, the quality of the observed data may have been adversely affected. However, the results obtained were satisfactory and classified as strong for the N-CE and N-RN areas on a seasonal basis (except for the SON season). The lower Pearson correlation coefficient found for the SON season, which experiences the highest wind speeds in the region, is linked to the limitations of reanalysis in representing extreme values of atmospheric variables. Reanalysis data provide the average of the variable over a time interval, impacting the estimation of maximum and minimum values. Maximum values tend to be underestimated compared to in situ measurements [13]. Studies by [58] also emphasize that the largest errors in reanalysis occur at stations with the highest average wind speeds.

The bias ranged between -0.5 and 0.5m/s for most of the year, successfully capturing the seasonal spatial pattern. This makes it suitable for use as a reference in validating the climate of the recent past period of the ECMWF-ERA5 reanalysis data when compared to the observed data. This spatial validation was important because the ECMWF-ERA5 reanalysis provides dynamic data with wind speed outputs at 100 m height, which is ideal height for our study as current wind turbines are approximately this height.

The seasonal validation between the RCMs and the ECMWF-ERA5 reanalysis data indicated that the RCMs can capture the wind speed patterns for the NEB region. However, they tend to overestimate or underestimate values for most of this region, leading to uncertainties in the evaluation of wind speeds in future scenarios. These discrepancies can be attributed to the inherent uncertainties in RCMs simulations. According to [18], bias errors can be characteristic of models, due to the complex, dynamic, and non-linear nature of the climate system, where natural variabilities can affect the entire system, thereby introducing uncertainties into the models.

Additionally, each RCM examined in this study is influenced by large-scale forcings from various GCMs, meaning that the parameters and physical input parameterizations from these GCMs contribute to the differences in results among the RCMs. [59] suggests that the quality of RCMs is inherently linked to the data derived from GCMs, encapsulated in the principle of "garbage in, garbage out". [60] argues that the empirical constants used in a model's physical parameterizations are based on the observations from the domains in which they were originally developed, introducing potential uncertainties when applied to different climatic regions. Therefore, for more specific studies, it is imperative to identify the most accurate inputs for the NEB region.

Another point to highlight regarding the discrepancies observed in the models relative to the reanalysis data pertains to the depiction of NEB's topography, which encompasses extensive regions of intricate terrain. The spatial resolution of the models (mostly within 50 km) may not adequately capture the nuanced topographical features of such complex areas. Additionally, it is pertinent to note that the wind speed outputs generated by the RCA4 and Remo2009 models were derived using logarithmic equations. According to [8], this method introduces uncertainties due to its dependence on surface roughness.

Regarding atmospheric phenomena, the ITCZ is characterized by a broad area of atmospheric instability and the formation of extensive convective clouds along the equatorial region [61]. This atmospheric system influences the northern portion of the NEB (when it is shifts further south) especially during the austral autumn (MAM). This southward intensification of

the ITCZ is more evident in the models: RegCM4.7_HadGEM2 (Fig 5A), RCA4_HadGEM2 (Fig 5D), Remo2009_MPI (Fig 5I), which overestimated wind speed in the northern region of the NEB during DJF and exhibited a reduction in both the bias intensity and the overestimated area during MAM, compared to the reanalysis data. Models RegCM4.7_NorESM, RCA4_NorESM and RCA4_Miroc (Fig 5C, 5F and 5H) depicted the ITCZ positioned further north relative to the MAM reanalysis, indicated by a decreased area and intensity of underestimation compared to DJF.

The obtained results show that the GCMs MPI and NorESM exhibited similar patterns when nested within different RCMs. The models RegCM4.7_MPI, RCA4_MPI and REMO2009_MPI displayed consistent wind speed patterns across all seasons. A similar consistency was observed for the RegCM4.7_NorESM and RCA4_NorESM models. The GCM, referred to here as EARTH, when coupled with the RCM RCA4 reproduced wind speed patterns similar to those of the GCM MPI (RegCM4.7_MPI, RCA4_MPI and Remo2009_MPI). The GCM Miroc, when combined with the RCM RCA4, showed wind patterns similar to the GCM NorESM (especially the RCA4_NorESM). This higher similarity may be associated with the fact that the parameters used in the RCM are the same. The models driven by the GCM HadGEM2 displayed different patterns, resembling each other only in the austral spring. These outcomes underscore the significant influence of GCM input data on RCM output results, as RCMs driven by the same GCMs tend to exhibit similar results. These findings corroborate those of [59].

In this study, nine sets of output data from three RCMs were analyzed individually. The bias results ranged from -2 to 2 m/s, with some regions exceeding this range. These findings are consistent with those of [10], who employed an ensemble of models using arithmetic means to reduce uncertainties and compare with the Era-Interim reanalysis data for the recent past period (1979–2005). Therefore, despite not using an ensemble of models in this research, the results are satisfactory as they fall within same margin of error. Generally, the models tended to overestimate wind speeds in the NEB region during most of the JJA and SON seasons, which are the months with the highest potential for wind power production. Consequently, when making future projections using these models, values may be overestimated compared to the reanalysis data.

The seasonality of wind speeds across selected areas in NEB is well-defined, with average velocities increasing in April, peaking in September and then declining. This seasonality plays a very crucial role in the electricity production for the NEB region. As illustrated in Fig 8A–8D, wind speeds intensify during the JJA and SON seasonal periods.

The highest annual average speeds associated with the N-CE and N-RN (Fig 8A and 8B) areas align with [6] due to their coastal locations, which present fewer barriers to wind. Additionally, the high variability observed throughout the year is linked to the migration of the ITCZ, which has a stronger influence in the N-CE area. The farther south the ITCZ is (MAM), the lower the associated wind speeds; conversely, the farther north the ITCZ is (SON), the higher wind speeds in this area [62]. In Borborema area (Fig 8C), the easterly winds can also contribute to a high average wind speed, with an annual average exceeding 6.0 m/s and the lowest standard deviation [63]. The wind speed annual average in C-BA is the lowest, at 5.04 m/s, which could be associated with the high and flat topography, as well as surface roughness that decelerates the winds [63]. Wind speed increase when the South Atlantic Subtropical High (SASH) moves to lower latitudinal positions (during the austral winter), this is also when C-BA experiences the highest wind speed [64]. In contrast, during the austral summer, this area has the lowest recorded wind speeds, possibly due to the passage of cold fronts [65].

Despite some states in the NEB experiencing drought during this period [5], there is an exception observed along the east coast of the NEB, including parts of the N-RN area (Fig 8B),

where the action of EWD [66] induces rainfall during the austral winter. This relationship between wind speed and precipitation is important for studying the hydro-wind complementarity in the NEB region. Moreover, during austral winter (JJA), the SASH shifts westward from the South Atlantic, suppressing convective activity and cold fronts in southeastern Brazil, leading to reduced precipitation [67]. Consequently, hydroelectric plants operate with low water levels, necessitating supplementary energy sources.

Wind energy serves as a clean and complementary source to hydroelectric power, underscoring its importance for the country's energy portfolio. Therefore, identifying areas with high potential for wind energy production becomes crucial. To archive this, it is important to minimize errors by selecting the RCM that best represents the wind speeds in the study area during the recent past from 1986 to 2005.

The RCMs were evaluated for the study areas, revealing persistent biases despite continuous efforts to improve their accuracy [10, 58, 68]. Furthermore, several studies have noted a high tendency for the models compared to reanalysis data [58, 69, 70]. The results from Taylor Diagram analysis were generally satisfactory for all study areas. All models showed correlation values above 0.7, indicating that they effectively capture the phases of the annual wind speed cycles in each region. However, the RMSE values suggest a tendency for similar simulation of the annual wind speed cycles by the models when compared to reanalysis. The RMSE values for the best models ranged from 1.02 m/s (C-BA) to 2.16 m/s (N-CE); these values probably arise from the high overestimation values for the JJA and SON seasons. Additionally, the Taylor diagram analysis highlighted that different models performed better in different areas, without a single model consistently demonstrating superiority across all NEB regions.

Therefore, it is important to acknowledge that the models exhibit significant uncertainties, and their performance can vary depending on their resolution and other factors. As highlighted by [10], the ensemble of RCMs used in studies across South America did not provide added value compared to GCMs when evaluated individually. Thus, calculating wind power density for the most significant wind energy-producing region using a single RCM or an ensemble may not always be the optimal choice. Instead, the preferred approach is to identify the model that best represents the observed data in a specific region of interest, as demonstrated in this study.

The study [71] evaluated RCMs for Fortaleza (CE) using precipitation accumulation data from the period 1970 to 2005. The authors found that the RegCM4.7_HadGEM2 model demonstrated good representativeness, ranking as the third best model assessed, with the top two models not evaluated in this study. Therefore, it was expected that this model would provide a better representation of wind speed compared to others in the N-CE area. Additionally, the RCA4_Miroc model, considered the best model in the Borborema and BA areas, was also regularly evaluated for precipitation in N-CE. This finding is significant, as the RegCM4.7_HadGEM2 model shows good representativeness for precipitation in the N-CE area, and our study confirms that it similarly represents wind speed well. Given the importance of these variables in this field of study, we can assert that the optimal approach for calculating wind power density in this area (N-CE) is using the RegCM4.7_HadGEM2 model.

Finally, the calculated values of wind power density have demonstrated that the NEB region is favorable for wind energy production. Understanding wind power density is important for identify areas where the wind speeds are sufficiently strong and consistent to generate electricity effectively and economically. This understanding also aids in energy demand planning, which can be met by this renewable energy source. The results indicate that between 59% and 80% of the wind power density corresponds to the second half of the year in Brazil, a period characterized by lower rainfall amounts. In this way, wind energy serves as a complementary source to hydroelectric power during this period. It is important to emphasize that for optimal

utilization of wind energy in the NEB, more comprehensive studies are necessary. These studies should consider factors such as the surface roughness of potential sites and the suitability of turbine types for each location.

## Conclusion

The objective of this research was to calculate the wind power density using Regional Climate Models in Northeast of Brazil (NEB) from 1986 to 2005, representing the recent past. This study is among the first to focus comprehensively on the NEB, comparing nine RCM with the newest ECMWF-ERA5 reanalysis data. It specifically examines areas with significant numbers of wind farms, identifying the best-performing models for each study area, analyzing wind speeds, and calculating wind power density for the recent past period.

In this research, the ECMWF-ERA5 reanalysis was validated against observed data, specifically the wind speed at 10m height. The results revealed a bias between -0.5 and 0.5m/s across most of the NEB for all seasons, with extensive regions showing moderate and strong correlation.

In the validation of the RCM participating in the CORDEX-CORE project, the models studied tended to either overestimate or underestimate wind speeds compared to the reanalysis at 100m height across all seasons in all NEB regions. There was significant overestimation of wind speed in the second half of the year, while errors were minimized during the first half, coinciding with the rainy season.

The behavior of wind speed at 100 m height was also analyzed. Coastal regions exhibited the highest average and maximum speeds, along with greater variability in wind speed, attributable to their geographical positioning and low soil roughness. All study areas delimited in the NEB are recognized as robust producers of wind energy, hosting the largest number of currently installed wind farms.

Among the RCMs evaluated for the NEB regions, those showing statistical metrics closest to the reanalysis were: RegCM4.7_HadGEM2 (N-CE and N-RN), RCA4-Miroc (Borborema and C-BA). This highlights that the dynamic downscaling technique enables RCMs to better represent certain areas over others, particularly in regions characterized by numerous climate phenomena.

Finally, wind power density was calculated for the recent past from 1986 to 2005. It was observed that the N-RN region exhibited highest wind potential, whereas the central region of Bahia showed the lowest potential among the studied areas. Coastal regions in Ceará and Rio Grande do Norte displayed more than double the wind potential compared to the other areas examined.

This work focuses on an approach to evaluating the wind speed of CORDEX RCMs based on seasonal averages (calculated from monthly means). It is worth highlighting that analysis on a smaller temporal scale (based on daily data) is also important, such as evaluating the trend of extreme wind events [72]. However, this was not the focus of this article. Some studies have already been conducted in Brazil [73, 74], but so far no studies have addressed this issue with a focus on the NEB region.

This research provides an analysis of wind speed behavior in this region, offering a scientific foundation for developing polices aimed at expanding wind power generation. Such initiatives contribute significantly to reducing GHG emissions. With effective strategic planning, it is anticipated that wind energy can grow substantially in the future, thereby reducing the country's reliance on hydroelectric sources and mitigating electricity generation challenges during drought periods.

## Author Contributions

**Conceptualization:** Augusto de Rubim Costa Gurgel, Domingo Cassain Sales, Kellen Carla Lima.

**Data curation:** Augusto de Rubim Costa Gurgel, Domingo Cassain Sales.

**Formal analysis:** Domingo Cassain Sales, Kellen Carla Lima.

**Investigation:** Augusto de Rubim Costa Gurgel, Domingo Cassain Sales, Kellen Carla Lima.

**Methodology:** Augusto de Rubim Costa Gurgel, Domingo Cassain Sales, Kellen Carla Lima.

**Project administration:** Augusto de Rubim Costa Gurgel, Kellen Carla Lima.

**Software:** Augusto de Rubim Costa Gurgel.

**Supervision:** Kellen Carla Lima.

**Validation:** Domingo Cassain Sales, Kellen Carla Lima.

**Visualization:** Domingo Cassain Sales, Kellen Carla Lima.

**Writing – original draft:** Augusto de Rubim Costa Gurgel.

**Writing – review & editing:** Domingo Cassain Sales, Kellen Carla Lima.

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
