## [Decision Letter · Decision Letter 0]

17 Mar 2024

PONE-D-24-02304Wind power density in areas of Northeastern Brazil from regional climate models for a recent pastPLOS ONE

Dear Dr. Gurgel,

Thank you for submitting your manuscript to PLOS ONE. After careful consideration, we feel that it has merit but does not fully meet PLOS ONE’s publication criteria as it currently stands. Therefore, we invite you to submit a revised version of the manuscript that addresses the points raised during the review process.

We look forward to receiving your revised manuscript.

Kind regards,

Delei Li, Ph.D.

Academic Editor

PLOS ONE

Journal Requirements:

Reviewers' comments:

Reviewer's Responses to Questions

**Comments to the Author**

1. Is the manuscript technically sound, and do the data support the conclusions?

Reviewer #1: Yes

Reviewer #2: Partly

2. Has the statistical analysis been performed appropriately and rigorously? 

Reviewer #1: Yes

Reviewer #2: No

3. Have the authors made all data underlying the findings in their manuscript fully available?

Reviewer #1: Yes

Reviewer #2: Yes

4. Is the manuscript presented in an intelligible fashion and written in standard English?

Reviewer #1: Yes

Reviewer #2: No

5. Review Comments to the Author

Reviewer #1: The article presents a compelling objective that addresses a critical need in the region, and if published, it will significantly contribute to the knowledge and applications of renewable energy sources in the area.

However, the abstract language could be improved by using and defining appropriate terms relevant to the study. For example, the phrase "data observed from Xavier" is ambiguous. Is it data observed by Xavier or observed data obtained from Xavier? Similarly, the phrase "but overestimated this variable" lacks clarity and coherence. I encourage the authors to revise their abstract to make it more concise and coherent.

In Figure 1, the unit of the topography is not specified. Additionally, the topography map does not indicate areas where the elevation is up to 2000. I suggest redrawing the map with a scale that ranges from 0 to 1500 at intervals of 200 to provide a more accurate representation of the topography.

I recommend providing a more detailed description of the Xavier data rather than just a reference. This section should give a general overview of the data. Similarly, a detailed description and appropriate references are missing in the ECMWF-ERA5 Reanalysis and CORDEX-Core data section. This should be updated as appropriate. It is important to include the physical parameterizations used in the RCMs in Table 1.

The bias in Figures 2 and 4-7 would be better represented with a colorbar that is not white-centered and has a smaller range, such as -1 to 1. This would provide a more accurate and vivid representation of the bias.

Reviewer #2: Wind power density in areas of Northeastern Brazil from regional climate models for a recent past

Manuscript PONE D-24-02304

Overview

The purpose of the work is to verify the skill of reanalysis and regional climate models to reproduce wind characteristics in Northeastern Brazil, and then to evaluate wind power density in different areas. Although the topic itself is of great interest in many aspects, the paper fails in some points. First of all, the English needs a deep review, by a professional reviewer or a native speaker, because some expressions are not usual and sound very strange. Secondly, the arguments for assuming a reasonable representation are not so clear and the figure quality needs to be improved. Discussions and conclusions are related to the results, but they are a bit vague and general.

In my opinion, rejected manuscript and suggestion to re-write and re-submit.

Specific comments

Please find my suggestions/comments in the balloons of the attached PDF using Acrobat Reader. If there is any problem to access them, please let me know.

6. PLOS authors have the option to publish the peer review history of their article (what does this mean?). If published, this will include your full peer review and any attached files.

Reviewer #1: No

Reviewer #2: No

---

## [Author Response · Author response to Decision Letter 0]

8 May 2024

Wind power density in areas of Northeastern Brazil from regional climate models for a recent past

Answers

Reviewer #1

It should be noted that the lines refer to the text with the corrections, that is, file revised_manuscript_with_track_changes_Augusto_plos_one.docx

Reviewer 1 correction (Line 29 up to 30) - However, the abstract language could be improved by using and defining appropriate terms relevant to the study. For example, the phrase "data observed from Xavier" is ambiguous. Is it data observed by Xavier or observed data obtained from Xavier? 

Answer: 

Thanks! The sentence has been modified to: “was validated in relation to the observed data obtained from Xavier”

Reviewer 1 correction (Line 35) - Similarly, the phrase "but overestimated this variable" lacks clarity and coherence. I encourage the authors to revise their abstract to make it more concise and coherent.

Answer: 

Thanks! The sentence has been modified to: Then, the Regional Climate Models RegCM4.7, RCA4 and Remo2009 were validated with the ECMWF-ERA5 reanalysis, which managed to represent well the wind speed pattern, principally from December to May. 

Reviewer 1 correction - In Figure 1, the unit of the topography is not specified. Additionally, the topography map does not indicate areas where the elevation is up to 2000. I suggest redrawing the map with a scale that ranges from 0 to 1500 at intervals of 200 to provide a more accurate representation of the topography.

Answer: 

Thanks! New figure with topography from 0 to 1200m, at intervals of 200m.

Reviewer 1 correction (Line 176 up to 183) - I recommend providing a more detailed description of the Xavier data rather than just a reference. This section should give a general overview of the data. 

Answer: 

Thanks! More details were provided from Xavier et al. (2016). The sentence has been modified to: “This database contains multiple atmospheric variables: precipitation, evapotranspiration, maximum and minimum temperature, solar radiation, relative humidity, and wind speed. The period of the database extended from 1980 to 2013. Six methodologies were employed for interpolating atmospheric variables, with the Inverse Distance Weighting (IDW) method emerging as the most effective for wind speed interpolation. The interpolation utilized databases derived from 3,625 pluviometers and 735 meteorological stations, collected by the National Institute of Meteorology (INMET), the National Water Agency (ANA), and the São Paulo State Department of Water and Electric Energy (DAEE) [22]”.

Reviewer 1 correction - Similarly, a detailed description and appropriate references are missing in the ECMWF-ERA5 Reanalysis and CORDEX-Core data section.

Answer: 

Thanks! The references were added and more description too. The sentence has been modified to:

In ECMWF-ERA5 (Line 198 up to 209)

“The monthly reanalysis ECMWF-ERA5 data was available in Copernicus Climate Data Store [13]. The ECMWF-ERA5 is a global reanalysis produced by the European Centre for Medium-Range Weather Forecasts (ECMWF), which features several improvements over its predecessor, the ERA-Interim. In terms of grid spacing, the ECMWF-ERA5 has a horizontal resolution of 31 km, compared to the 80 km of the ERA-Interim. Moreover, ECMWF-ERA5 includes an enhanced vertical resolution and hourly output, in contrast to the six-hourly output of the ERA-Interim. Additionally, the ECMWF-ERA5 incorporates a 4D-Var data assimilation system, representing an advancement over the ERA-Interim. Regarding physical parameterizations, ECMWF-ERA5 integrates the HTESSEL land surface scheme, which has seen significant improvements over the ERA-Interim, including an enhanced representation of hydrology, the introduction of soil texture maps, and improved snow cover parameterization. Furthermore, ECMWF-ERA5 benefits from the ongoing development in all-weather condition assimilation, including the successful extension to microwave humidity data [13].”

In RCMs (Line 223 up to 227)

“This study was performed by three RCMs from CORDEX-CORE [16]. The RegCM4.7 was nested in three GCMs outputs (HadGEM2-ES, NorESM1-M and MPI-ESM-MR). The RCA4 was nested in five GCMs outputs (HadGEM2-ES, NorESM1-M, MPI-ESM-LR, EC-EARTH and Miroc5), while Remo2009 was nested only in MPI-ESM-LR GCM output. This combination of regional-global models results in nine output data.”

Reviewer 1 correction (Line 228 up to 241) - This should be updated as appropriate. It is important to include the physical parameterizations used in the RCMs in Table 1.

Answer: 

Thanks! The physical parameterizations were added. The sentence has been modified to: 

“The RegCM4 model utilizes various parameterizations to simulate atmospheric and surface processes. The radiation scheme is the NCAR CCM3 [23]. The land surface model scheme employed in RegCM4 is the Biosphere-Atmosphere Transfer Scheme (BATS) [24], while the cloud microphysics description used is based on the European Centre for Medium-Range Weather Forecasts' Integrated Forecast System (IFS) [25,26,27]. Furthermore, the planetary boundary layer scheme used in the RegCM4 model is the Holtslag planetary boundary layer scheme [28].

 Regarding the RCA4 model, the parameterization for radiation is based on the HIRLAM radiation scheme, originally developed for numerical weather prediction purposes [29]. For the surface, the RCA4 model utilizes the BATS surface scheme [30], while the surface scheme used in the model is referred to as the Land-Surface Scheme (LSS) and belongs to the second generation of LSSs [31]. The Remo2009 model features a physical radiation parameterization known as the Morcrette radiation scheme, which is utilized in the ECHAM4 general circulation model [32]. Furthermore, the model incorporates a cloud microphysics scheme named PCI [33], and [34] have developed a global dataset of land surface parameters (LSP).”

Reviewer 1 correction - The bias in Figures 2 and 4-7 would be better represented with a colorbar that is not white-centered and has a smaller range, such as -1 to 1. This would provide a more accurate and vivid representation of the bias.

Answer: 

Thanks! The figures have been redone. The range was reducted to -2 to 2 because the only studied to South America has this range. So, it is important to compare the results obtained with previous studies such as Reboita et al. (2018).

Reviewer #2

Reviewer 2 correction - First of all, the English needs a deep review, by a professional reviewer or a native speaker, because some expressions are not usual and sound very strange. 

Answer: 

Thanks! The text was reviewed. 

Reviewer 2 correction - Secondly, the arguments for assuming a reasonable representation are not so clear and the figure quality needs to be improved. 

Answer: 

Thanks! The arguments were rewritten. The figures quality was improved. The figure sent in the first version of the article in the .tif extension presented the resolutions desired by the journal, but when added to the .pdf there was a loss of quality.

Reviewer 2 correction - Discussions and conclusions are related to the results, but they are a bit vague and general. 

Answer: 

The argument below is not in the article. 

The first discussion in this study is concern about the choice of the ECMWF-ERA5 reanalysis. This choice is based on studies demonstrating its superiority over other reanalysis, including compared to the Era-Interim reanalysis used by Reboita et al. (2018) in a similar study for the same study area.

Subsequently, the reanalysis is assessed against observational data. Spatial evaluation of reanalysis against observed data is not commonly conducted in articles, and when such evaluations are performed, they are often done on a point-by-point basis. This represents a unique aspect of our study, as there are no established benchmarks in the literature for comparing our results with those of other studies. This approach aids in understanding how the reanalysis, widely used as observed data in studies in this field, represents observed data in Northeast Brazil.

Next, models are assessed in relation to the reanalysis, where it is common to encounter errors that are frequently under-discussed in articles. This article aims to consolidate diverse potential sources of error, covering both model uncertainties and analyses derived from atmospheric systems, among others. The identified errors are consistent with the findings of Reboita et al. (2018), enhancing studies for this region, which are limited in the context of utilizing regional climate models.

Finally, the seasonality of wind speed in the Northeast is evaluated and explained through atmospheric systems. Models are assessed using Taylor diagrams for regions with the highest concentrations of wind farms. The model that best approximates the reanalysis is selected based on statistical metrics. This is crucial because, for example, in the study of precipitation in the city of Fortaleza (CE) located in Northeast Brazil, one of the top-performing models chosen to represent precipitation was the RegCM4.7-HadGEM2, similar to our findings regarding wind speed. This is important because accurately representing these two variables can provide insights into other variables and thus simulate future scenarios with less uncertainty (an objective of a forthcoming study). 

Lastly, the models with the lowest uncertainties were used to calculate wind power density for the past. Despite such results being included in atlas, it generally do not calculate wind power density using models and therefore cannot simulate the future. This article opens up opportunities to compute wind potential for the future, which is significant for the academic community.

From here on, the responses will be according to the PDF.

Reviewer 2 correction (Lines 32 up to 35) - Specific comments: Not so high correlation values, which means that the variability is not so well represented...

Answer: 

Thanks! “The results were satisfactory, with strong correlation in areas of Ceará and Rio Grande do Norte (except in SON season), and some differences in relation to the wind intensity registered by observed data, mainly for JJA season.” - the phrase was changed. 

The argument below is not in the article. 

Strong correlation in our specific areas, particularly in Ceará and Rio Grande do Norte, which exhibit the highest wind power density, has been observed. The reanalysis chosen is supported by numerous articles demonstrating its superiority over others, for instance, ERA5 outperforms Era-Interim, as used in Reboita et al. (2018) study in the same area. This represents the first study utilizing this observed database for the reanalysis. Further studies are needed to address why the variability is not well represented.

Reviewer 2 correction (Line 46) - “this can be found in wind atlas, and it's not new”

Answer: 

The argument below is not in the article. 

This topic was argued above: the atlas calculates wind power density based on observed data. This study calculates wind power density using selected models in areas with the highest concentrations of wind farms and the lowest uncertainties relative to the ECMWF-ERA5 reanalysis. This enables the projection of wind power density into the future, which is not seen in wind atlas. Future projection is part of the objectives of a forthcoming article.

Reviewer 2 correction (Line 160) - what is this? this information should add something to the reader?

Answer: 

Thanks! The information was removed. “Borborema (yellow): the region of the Borborema plateau that covers the states of Rio Grande do Norte, Paraíba and the wild landscape (agreste) of Pernambuco, and finally, C-BA (green): comprises the region of central Bahia.”

Reviewer 2 correction (Line 177) - atmospheric

Answer: 

Thanks! The word was modified: “This database contains multiple atmospheric variables”

Reviewer 2 correction (Line 189 up to 190) - interpolated at horizontal grid points of high resolution 0.25°×0.25° this is a characteristic of the dataset and was not performed by the present authors my suggestion is to re-write the description of the dataset.

Answer: 

Thanks! The sentence has been rewritten “The authors utilized wind speed data at 2m intensity for the period from 1986 to 2005, which was obtained from a dataset with a horizontal resolution of 0.25°×0.25° latitude-longitude covering all of Brazil. This high-resolution dataset was employed as part of the study, but the interpolation process itself was not conducted by the present authors”

Reviewer 2 correction (Line 210) - 10 m from the ground. why not "100 m height"?

Answer: 

Thanks! Correction accepted and incorporated throughout the entire article. “10 m from the ground height”

Reviewer 2 correction (Line 211) - used to validate the observed data to evaluate ERA5 performance against observed data .it is not the observations that are being validated...

Answer: 

Thanks! Correction accepted: “used to evaluate ERA5 performance against observed data.”

Reviewer 2 correction (Line 277 up to 282) - does this extrapolation consider atmospheric stability?

Answer: 

Thanks! The explanation was added: “ Eq ([Disp-formula pone.0307641.e001]) is appropriate for neutral stability conditions of the atmosphere. In the NEB, sea winds are expected to present neutral to unstable atmospheric conditions, as demonstrated by climatological data [45] and radiosonde observations [46]. Furthermore, in [10], the equation was employed without separating stability conditions because the study dealt with monthly data and a climate scale, rather than at a micrometeorology scale, similar to the situation in this study.”

Reviewer 2 correction (Line 273-274) - height of the die relative to the ground

Answer: 

Thanks! The sentence has been rewritten “ Zr = the height of the die reference point (ex: meteorological station) relative in relation…”

Reviewer 2 correction (Line 286 up to 287) - time series of 20 points for statistics are quite short

Answer: 

The sentence was rewritten: “This validation occurred seasonally (with 60 values for each season, covering a period of 20 years) with the wind speed variable at 10 m height.”

Reviewer 2 correction (Line 297) - does this word exist?

Answer: 

Thanks! The word was rewritten: The word not exist “wherel”, correct is “where”

Reviewer 2 correction (Line 318 up to 319) - table 2 could be removed

Answer: 

Thanks! The table was removed and the citation was retained in the text

Reviewer 2 correction (Line 328) - boxplots are monthly

Answer: 

Thanks! The word was corrected “In this analysis, boxplots were constructed to examine monthly.

Reviewer 2 correction (Line 342 and 343) - considering the area averages and monthly means, how the authors can say "with the least possible error"?

Answer: 

(1) The expression was rewritten : “This validation process facilitates the calculation of wind power density in these areas with reduced uncertainty [18].”

(2) This answer is not in the Article - The objective of the study is to calculate the wind power density with the best RCM in the chosen area from NEB. This study considers a climate scale with monthly data rather than a micrometeorology scale. So, for this objective the area average and monthly means could be able to reduce the uncertainty. Considering the areas and a high resolution of the RCMs the results could reduce the uncertainty. For example, inside of the chosen areas the climate and the wind speed are very similar. In this study we consider only the wind speed value. Considering climate scale, for example, for monthly means the wind speed value is similar for the east and west in the RN area, so the area average could be a good representation. Another point is that the unique spatially study for wind speed in South America is by Reboita et al. (2018). This study will contribute to expanding the literature. Authors are engaged to write new articles, using daily data in the future.

Reviewer 2 correction (Line 368 up to 370) - Stimulated e observed 

Answer: 

Thanks! The words was rewritten: “Where: Pdata,i is the simulated value o

---

## [Decision Letter · Decision Letter 1]

28 May 2024

PONE-D-24-02304R1Wind power density in areas of Northeastern Brazil from regional climate models for a recent pastPLOS ONE

Dear Dr. Gurgel,

Thank you for submitting your manuscript to PLOS ONE. After careful consideration, we feel that it has merit but does not fully meet PLOS ONE’s publication criteria as it currently stands. Therefore, we invite you to submit a revised version of the manuscript that addresses the points raised during the review process.

We look forward to receiving your revised manuscript.

Kind regards,

Delei Li, Ph.D.

Academic Editor

PLOS ONE

Journal Requirements:

Reviewers' comments:

Reviewer's Responses to Questions

**Comments to the Author**

1. If the authors have adequately addressed your comments raised in a previous round of review and you feel that this manuscript is now acceptable for publication, you may indicate that here to bypass the “Comments to the Author” section, enter your conflict of interest statement in the “Confidential to Editor” section, and submit your "Accept" recommendation.

Reviewer #2: (No Response)

2. Is the manuscript technically sound, and do the data support the conclusions?

Reviewer #2: Yes

3. Has the statistical analysis been performed appropriately and rigorously? 

Reviewer #2: Yes

4. Have the authors made all data underlying the findings in their manuscript fully available?

Reviewer #2: Yes

5. Is the manuscript presented in an intelligible fashion and written in standard English?

Reviewer #2: Yes

6. Review Comments to the Author

Reviewer #2: Although my suggestions were incorporated into the manuscript and the English was improved, the text still needs some language verification and some discussion/explanation about the following topic:

-> is the variability not important for a wind farm? This question was firstly posed in the context of the use of monthly data information against high-frequency variability. A paragraph on that could be included in the discussion section. Please note that this topic can also be directly related to a second aspect mentioned in the cover letter and included into the text: "Reanalysis provides the average of the variable over a time interval, impacting the estimation of maximum and minimum values.", which is related to the relative low performance of ERA5 in SON. So, there is some missing variability in the high frequency as well as in the monthly averages. How do manage with that when considering future projection, for instance?

Moreover, equation 8 is not correct, simply because the denominator should be (273.15 + T), I mean, the "T" is not added to the prior fraction, being is part of its denominator. Please correct.

7. PLOS authors have the option to publish the peer review history of their article (what does this mean?). If published, this will include your full peer review and any attached files.

Reviewer #2: No

---

## [Author Response · Author response to Decision Letter 1]

18 Jun 2024

Wind power density in areas of Northeastern Brazil from regional climate models for a recent past

Answers

Reviewer #2

1) Although my suggestions were incorporated into the manuscript and the English was improved, the text still needs some language verification.

Answer: Thanks! The English was improved. If needs more improve, please select the sentences. 

2) some discussion/explanation about the following topic:

-> is the variability not important for a wind farm? This question was firstly posed in the context of the use of monthly data information against high-frequency variability. A paragraph on that could be included in the discussion section. Please note that this topic can also be directly related to a second aspect mentioned in the cover letter and included into the text: "Reanalysis provides the average of the variable over a time interval, impacting the estimation of maximum and minimum values.", which is related to the relative low performance of ERA5 in SON. So, there is some missing variability in the high frequency as well as in the monthly averages. How do manage with that when considering future projection, for instance?

Answer: Thanks! A paragraph was written in conclusion section.

3) Moreover, equation 8 is not correct, simply because the denominator should be (273.15 + T), I mean, the "T" is not added to the prior fraction, being is part of its denominator. Please correct.

Answer: The equation was corrected

---

## [Editor Report · Decision Letter 2]

9 Jul 2024

Wind power density in areas of Northeastern Brazil from regional climate models for a recent past

PONE-D-24-02304R2

Dear Dr. Gurgel,

We’re pleased to inform you that your manuscript has been judged scientifically suitable for publication and will be formally accepted for publication once it meets all outstanding technical requirements.

Kind regards,

Delei Li, Ph.D.

Academic Editor

PLOS ONE
---

## [Editor Report · Acceptance letter]

16 Jul 2024

PONE-D-24-02304R2 

PLOS ONE

Dear Dr. Gurgel, 

I'm pleased to inform you that your manuscript has been deemed suitable for publication in PLOS ONE. Congratulations! Your manuscript is now being handed over to our production team.

Kind regards, 

on behalf of

Dr. Delei Li 

Academic Editor

PLOS ONE